# Digitally Enabled Experiential Learning Spaces for Engineering Education 4.0

David Ernesto Salinas-Navarro [1,*], Claudia Lizette Garay-Rondero [2,*] and Iván Andrés Arana-Solares [2]

1    Aston Business School, Aston University, Birmingham B4 7ER, UK
2    Institute for the Future of Education, School of Engineering and Sciences, Tecnologico de Monterrey, Ave. Eugenio Garza Sada 2501, Monterrey 64849, Mexico
*    Correspondence: d.salinas-navarro@aston.ac.uk (D.E.S.-N.); clgaray@tec.mx (C.L.G.-R.)

**Abstract:** Novel digital technologies have transformed societies, organizations, and individuals in diverse aspects of daily life, elevating their competency requirements in order to successfully develop, integrate, and generate value. To remain relevant Higher education should provide students with digitally enhanced learning experiences to build their necessary competencies. To progress in this direction, this work proposes a method that can be used to develop *digitally enabled experiential learning spaces* (DeELS) in engineering education so as to incorporate digital technologies into engineering problem-solving and decision-making activities, as an innovative approach to Education 4.0. Two implementation cases exemplify the digital transformation of these learning spaces in the Lean Thinking Learning Space (LTLS) for undergraduate engineering courses. The exemplification shows how students, through designing, creating and integrating digital/smart kanban systems, execute their learning activities in a DeELS. The results suggest that the students were able to satisfactorily achieve their learning outcomes through the learning experiences. Moreover, new instances of learning experiences for digital transformation were identified within the LTLS. However, future work is required regarding new instances of digital transformation learning experiences in order to make any further inferences or generalizations regarding DeELS and their contribution to competency development.

**Keywords:** education 4.0; engineering education; experiential learning spaces; educational innovation; higher education

## 1. Introduction

In recent years, digital technologies have revolutionized manufacturing engineering and production techniques. This revolution, popularly called Industry 4.0, has resulted from the technological integration of cyber-physical systems, the internet of things, computing technologies, automation, robotics, and autonomous vehicles and their incorporation into industrial applications that enhance manufacturing, production, and operation performance [1]. Moreover, societies, organizations, and individuals have embraced diverse internet-based technologies for online communication, data collection and analysis, and geo-localization over recent decades. These technologies include artificial intelligence, augmented/virtual reality, predictive analytics, ground positioning systems, and others which have been used in various activities, fields, and people's lives in what has been called the digital transformation [2].

Industry 4.0 and digital transformation require the technical development and incorporation of digital technologies in diverse systems, processes, and operations [3,4]. There is also the need to integrate digital technologies into products and services, value chains, and business models [5–7]. For people, technological change affects labor markets and the future of jobs [8]. This situation demands specialists with the digital skills necessary to reach digital transformation in organizations and assume more complex job positions and business opportunities.

Accordingly, educating new professionals about the technological changes in Industry 4.0 and digital transformation presents significant challenges. The World Economic Forum (WEF) defines Education 4.0 as referring to the preparation of the next generation of talent through relevant tools and insights in order to face current societal challenges, closing the existing education gap in technological accelerators [9]. Thus, Education 4.0 refers to teaching the technical aspects of digital technologies and developing the necessary abilities to lead, manage, deploy, operate, and act in the digital change environment [9]. This type of education is particularly relevant to engineering education because it implies developing the capacity of students to engineering digital solutions in their future professional careers [10]. In the case of industrial engineering education, one of the disciplines that is focused on this work, this idea means that students should learn to integrate digital technologies into the management and production process, with the aim of improving the delivery of products and services [11].

However, learning to engineer solutions of this type requires changes in, or adaptations of, the existing educational approaches. The presence of technology influences learning and the spaces in which this happens, as there is a close connection between the two and reciprocal impacts in enhancing, enabling, and extending each other's scopes, functions, and roles [12–14]. Hence, educational approaches and learning spaces must be adapted accordingly [15]. Concerning learning spaces for Education 4.0, these should allow students to engage in active and reflective hands-on learning and integrate internet connectivity with an appropriate architectural environment and innovative furniture, gadgets, tools, and equipment [16,17]. Some examples related to engineering education can be found in the integration of statistical analysis software [18], simulations, open-access databases, mobile apps, and geographical information systems into learning spaces [19]. In these cases, students use information and communication technologies in a classroom or lab to enhance their learning or explore challenging learning situations. Nevertheless, digital technologies might also be used as part of immersive learning experiences, in which students undertake problem solving and decision making in contrived learning environments or real-world situations [20,21]. That is, there is an identified need in Education 4.0 to learn by doing in immersive, digitally enabled learning spaces for experiential learning [10,22]. Current approaches to industrial engineering education in Industry 4.0 highlight the need for, and provide examples of, learning activities aimed to develop digital competencies concerning quality improvement [23], project management [24], human factors [25], logistics [26], and user experience [20]. However, there is no reference to learning spaces or the educational infrastructure necessary for experiential learning in engineering education and Education 4.0 for this purpose.

This proposition regards the innovations necessary for challenging Industry 4.0 and digital transformation learning activities in *experiential learning spaces* for engineering education, specifically industrial engineering, that contribute toward Education 4.0, namely Engineering Education 4.0. Therefore, this work aims to (i) offer a framework with which to develop Education 4.0 learning experiences in experiential learning spaces and (ii) a method with which to implement this framework, and (iii) to report on the exemplification of these ideas in undergraduate engineering courses.

Accordingly, this manuscript presents five additional sections. Section 2 presents a literature review of experiential learning and the relationship between digital transformation and engineering education, as well as the need to develop digital competencies to face the current challenges in a global society. Section 2 also presents a method and an Engineering Education 4.0 framework with which to develop innovative learning activities within digitally enhanced experiential learning spaces. Additionally, Section 2 provides the methodology that can be applied to explore the use of this method, aiming to develop digitally enabled experiential learning spaces for Engineering Education 4.0. Section 3 describes the results of two instances of digital transformation implementations within the *Lean Thinking Learning Space*, as application cases, providing us with the opportunity to observe and collect data for an analysis of the challenging learning experiences in a lean

manufacturing situation. Section 4 presents the analyses and discussion of the implementations and the main findings of this work. Finally, Section 5 concludes the discussion with this research work's main contributions and future possibilities.

## 2. Materials and Methods

### 2.1. Experiential Learning Spaces in Engineering Education

Digital transformation in the 21st century demands creative and innovative abilities to face the wave of new technologies, disruptive business models, and changing divisions of labor between workers and machines [8]. This situation changes the demand for physical and manual skills, favoring digital abilities that address the range of skills required in all aspects of professional life, both now and in the future. This assertion focuses on the development of digital literacy and a series of competencies and sub-competencies defined as *digital transformation competencies*. Consequently, a gap exists in regard to what, how, and where to teach students about using, creating, and implementing new technologies. Hence, traditional learning spaces are no longer enough.

Digital transformation competencies encompass the knowledge of, abilities in, and attitudes toward digital technologies that people possess, as well as their effectiveness and critical thinking for the achievement of specific purposes [27,28]. These competencies include the technological, communicative, collaborative, informational, and multimedia aspects of complex literacy competency. However, the competencies also imply those skills necessary to create, develop, and apply digital technologies that generate value in professional disciplines and society [29]. Furthermore, the WEF articulates these requirements as an interplay of skills of global citizenship, innovation and creativity, digital technology, and interpersonal skills for Education 4.0, as well as educational experiences to foster personalized, accessible, inclusive, problem-based, collaborative, lifelong, and student-driven learning [30].

Engineering, as a discipline, involves the integration of problem solving and technological development. Therefore, engineering education should approach this goal holistically [31,32] and map complex problems in order to solve them in the learning process and to create learning experiences that meet the educational requirements. This idea implies adequate educational objectives, teaching strategies, learning spaces, and learning experiences, supporting students in achieving their expected competencies and learning outcomes [33]. Therefore, engineering education should adapt learning experiences to produce competencies aligned with digital transformation skill requirements [10]. *Learning experiences*, here, refer to a wide variety of events in which the learner transforms her/his perceptions, understanding, emotions, knowledge, skills, and attitudes [34].

Moreover, learning experiences for digital transformation entail changing the places where learning happens, that is, moving into learning spaces that support digital transformation [22,35]. *This new view calls for the digital transformation of learning spaces for Education 4.0.* Previous definitions or elaborations of learning spaces do not consider this notion of learning spaces.

A learning space is commonly referred to as a physical place, such as a classroom, laboratory, or workshop, where learning occurs [36]. Learning spaces align with the learning and teaching goals and the school's mission and integrate technology and educational resources to develop students' competencies [37]. Accordingly, Education 4.0 requires specific types of learning spaces.

To progress in this direction, one possibility can be found in experiential learning spaces [38–40]. Hence, *experiential learning spaces* refer to those spaces that support learners in their action/reflection and experience/conceptualization of a situation, individually or collectively.

At present, learning spaces should support situated, active, and reflective *learning by doing*, with real-life issues or problems [35]. If experiential learning is to occur, students should learn through an interplay of thinking and acting in a situation, aligning their learning purpose with their activities and practical experiences [41]. Hence, according to Kolb's



experiential learning cycle, the pedagogy underpinning this work, experiential learning spaces should promote specific active experiences to perceive, reflect upon, conceptualize, and experiment in order to achieve the intended objectives and learning outcomes [42–44]. Examples of experiential learning in engineering education can be found across disciplines, with all of them stressing the importance of the experiential learning cycle for developing engineering competencies and the learning outcomes of the curriculum [45–48].

From a systemic point of view, this work considers a learning space as a social domain of interaction, where the participants interact to achieve a specific learning purpose within a structure of activities, roles, and resources [33,41]. That is, learning spaces go beyond traditional notions to involve aspects of educational infrastructure and resources; social interactions and collaboration; means of contact, proximity, and integration; and the cooperation of participants over time. This novel proposition draws attention to the purposeful interactions of the participants rather than the required physical resources and infrastructure of a classroom alone. Hence, in engineering education, Education 4.0 requires us to consider digital technologies as an inherent resource in learning spaces, supporting the interactions of the participants and leading to experiential learning in a situation. This new concept of experiential learning spaces enables the further evolution of the notion. Nevertheless, suitable experiential learning spaces should be developed for Education 4.0 to meet their specific purposes.

Some authors have used diverse pedagogical approaches to acquire digital skills in different learning spaces for engineering education (see [33,49–54]). For instance, some works explored the development of digital skills in labs by having students carry out practical work with digital technologies [52–54]. Miranda et al. proposed a combination of pedagogical methods, information and communication technologies, and educational infrastructure for Education 4.0 [16]. Other researchers agreed that experiential learning provides the opportunity to teach new digital technologies and offer students an Industry 4.0 experience, without referring to the necessary learning spaces [20,48,49,51]. For instance, Bonavolontà et al. proposed a remote laboratory design and its implementation, enabling students to learn about automation focused on digital technologies [50]. These authors mentioned that developing a remote laboratory allows students to become familiar with the principles and technologies of Industry 4.0, following the experiential learning cycle.

Digitally enhanced experiential learning spaces intended for Education 4.0 in engineering education have scarcely been researched. That is, learning spaces that incorporate digital technologies as learning resources to support the experiential learning of Industry 4.0 and digital transformation are little understood. Despite the fact that authors such as Salinas-Navarro et al. and Garay-Rondero et al. have carried out studies of experiential learning spaces in lean manufacturing labs, there is still little literature on the subject, which this work addresses [10,55]. Referring to these previous works, the notion of experiential learning spaces concerns the integration of challenging experiential learning activities into an active, collaborative, and immersive physical space for lean manufacturing education. Additionally, these authors provided a conceptualization of the integration of 4.0 technologies into a learning space for a hands-on immersive production process in industrial engineering education, but no work was conducted regarding its implementation. This limitation justifies the present work, aiming to advance in the exemplification and implementation of this type of learning space and to collect data and study the educational contribution of such spaces to the development of Education 4.0 learning experiences.

### 2.2. Digitally Enabled Experiential Learning Spaces

The use of digital technologies has resulted in digital factories, smart operations, intelligent supply chains, the digitalization of products and services, and the digital transformation of business models [56,57]. Accordingly, Education 4.0 should occur in learning spaces that develop solutions to problems similarly, using digital technologies [10,33]. These

innovations and technologies must be integrated into an experiential learning process for Engineering Education 4.0 [58].

Regarding digital competencies in Engineering Education 4.0, experiential learning calls for a definition of digitally enabled experiential learning spaces (DeELS). Accordingly, the authors propose the following construct:

*DeELS are innovative learning spaces that produce relevant and challenging Industry 4.0 and digital transformation learning experiences based on experiential learning activities, in which digital technologies represent an educational resource for Education 4.0.*

Figure 1 shows a framework that can guide the development of experiential learning spaces for Education 4.0 in engineering education. The framework includes the transition from the external to the internal sections, (i) a collection of learning requirements in Industry 4.0 and digital technologies settings for digital literacy competencies [10]; (ii) components of educational design concerning the intended learning outcomes, digital competency development, and teaching and learning strategies and approaches [59]; (iii) learning spaces with respect to the social interactions, types of contact, infrastructure and resources, and time coincidence of the participants [33,41,60]; (iv) challenging learning experiences taking place in the learning space; and (v) experiential learning activities following Kolb's experiential learning cycle [44]. This is a generic framework that can be used to guide the recreation of DeELS across different engineering disciplines for Education 4.0. The relationship between Education 4.0 and DeELS is described in Table 1, pointing to the key characteristics of the learning outcomes and digital competency.

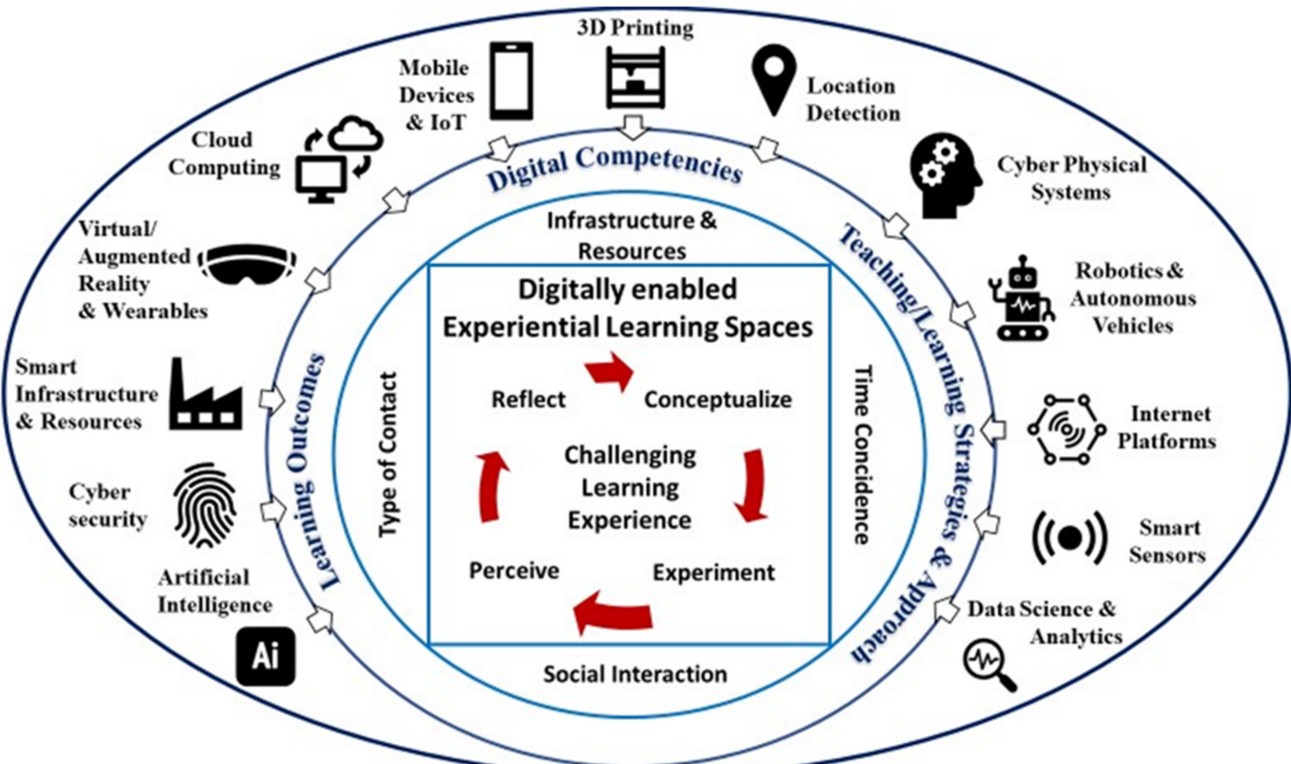

**Figure 1.** Digitally enabled experiential learning spaces framework (own elaboration) [44]. Requirements of digital technologies translated into educational components that define challenging learning experiences within a DeELS to develop digital competencies.

Next, a method that can be used to develop DeELS is presented to provide a step-by-step procedure with which to conceptualize instances, as follows:

1. Define the educational purpose as the objectives, learning outcomes, disciplinary and digital transformation competencies, pedagogical requirements, and learning experience characteristics of Education 4.0 [61].
2. Identify the experiential learning activities necessary to carry out the digital transformation of a learning space [10].
3. Conceptualize a purposeful DeELS [33,41].
4. Design challenging learning experiences within the DeELS [33,55].

**Table 1.** Education 4.0 and its relation to DeELS.

| Education 4.0 | Learning Outcome | Digital Competency | Experiential Learning Space Characteristics | DeELS Characteristics |
|---|---|---|---|---|
| Teaching the technical aspects of digital technologies and developing the necessary abilities to lead, manage, deploy, operate, and act in the digital change environment [9]. | Generate solutions to problems in the professional field, with the intelligent and timely incorporation of novel digital technologies [29]. | The ability to evaluate various digital technologies with openness to search for and implement relevant alter-natives in order to transform professional practice, considering economic, environmental, social, political, ethical, safety and hygiene, and manufacturing restrictions [29]. | - Innovative conceptualization of learning spaces. <br> - Support the development of the intended learning outcomes and competencies. <br> - Produce generic relevant and challenging learning experiences. <br> - Based on experiential learning activities. <br> - Use of diverse educational resources across different educational settings [33,38,41]. | - Innovative conceptualization of learning spaces, involving digital enablers. <br> - Support the development of Education 4.0 learning outcomes and competencies. <br> - Produce relevant and challenging Industry 4.0 and digital transformation learning experiences. <br> - Based on experiential learning activities for digital literacy competency in engineering education. <br> - Use of digital technologies as an educational resource for Education 4.0. |

The first step of the method is to declare the intentions, purposes, and structure of the learning activities and educational resources that actively develop the competencies of the students through challenging learning experiences. Referring to engineering education, in this case, this work considers a generic *digital learning outcome* for Education 4.0, concerning students' capacity for generating solutions to problems in the professional field, with the intelligent and timely incorporation of novel digital technologies. This definition is linked to *Education 4.0 digital competency regarding* "the ability to evaluate various digital technologies with openness to search for and implement relevant alternatives to transform professional practice, considering economic, environmental, social, political, ethical, safety and hygiene, and manufacturing restrictions" [29]. Other definitions of digital competencies might exist; however, this work focuses on the previously mentioned definition of competency in terms of the digital transformation of a DeELS.

Furthermore, in the second step, a digital transformation requires students to plan, reflect upon, design, and implement a digital device or system, following the experiential learning cycle within a learning space, working in multidisciplinary teams, and incorporating digital technologies.

Thirdly, specific implementations of digital technologies within a DeELS relate to the ability to create engineering solutions so as to improve production processes and operations and meet the demands of customers and educational partners, including enterprises, institutions, and organizations. Fourthly, a learning challenge definition must incorporate the specific digital transformation needs, use, and implementation of technological devices. A challenging design anticipates the leadership, culture, and organization necessary for digital transformation (such as the operators' training, teamwork, and problem-solving abilities). Other factors are customer experience and satisfaction, the contribution to business/organizational objectives, and the creation of products that meet market requirements. For instance, a DeELS should facilitate challenging learning experiences in industrial engineering using digital technologies to eliminate waste, improve quality acceptance and service levels, and maximize operational efficiency. Thus, this

method is exemplified in detail in Section 3 through two cases applying a DeELS to show its implementation and the development of specific learning experiences in undergraduate courses. With this knowledge, the notion of DeELS can be replicated and used in further educational applications.

*2.3. Methodology*

A methodology that can be used to develop a DeELS and report on its implementation as case studies is presented in this section. The methodology of this work unfolds in three stages based on the method presented in Section 2.2, as shown in Figure 2.

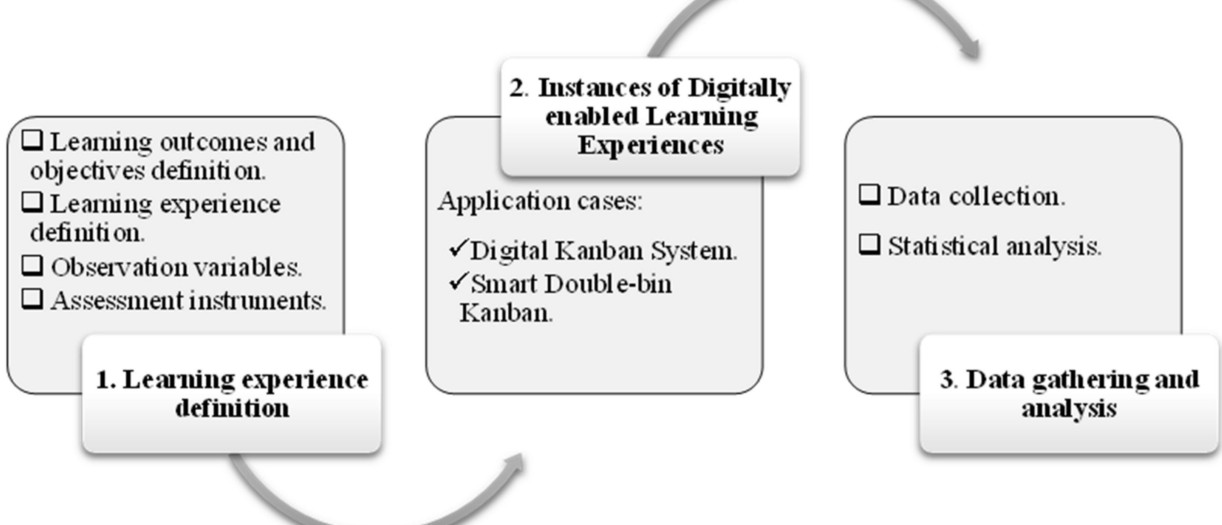

**Figure 2.** The methodology (own elaboration).

2.3.1. Stage 1

The first stage defines the learning experience using the method used to develop the DeELS (see Section 3.1), the observation variables, and the data collection instruments (explained later in this section). Table 2 describes the observation variables and instruments for data collection in regard to the learning experience, namely evaluation rubrics, checklists, and a student survey. Digital transformation competency is observed through assessments during the learning experiences. Students are expected to show their knowledge and skills during the execution of their learning activities, and instructors are expected to evaluate the students' performance and learning outcomes using an evaluation rubric and checklist (see Appendix A) [62,63].

**Table 2.** Observed variables of the challenging learning experience.

| Variable = X | Type | Response Variable = Y | Assessment Levels | Data Collection Instruments |
|---|---|---|---|---|
| Digital transformation competency | Ordinal qualitative | Self-assessment of students' achievement level | Level A = 3<br>Level B = 2<br>Level C = 1 | Competency achievement rubric and single-point checklist (Appendix A) |
| Numeric grades for project evaluation | Continuous quantitative | Learning outcomes evaluation | Score 1–100 points | Grading rubric (Appendix C) |
| Student's opinions | Ordinal qualitative | Students' opinions on the learning experience | Likert scale opinion (1–5) | Opinion survey (Appendix B) |

Moreover, the students' numerical grades (e.g., project reports) provide tangible evidence of their digital competencies, which are evaluated using an achievement level rubric (see Appendix B). The students should also report their perceptions of and opinions on their learning experience by answering questions in a survey about the relevance of their activities, their motivation and interest, and the learning experience's contribution to advancing their digital transformation skills (see Appendix C).

### 2.3.2. Stage 2

The second stage refers to the reporting of the results of application cases regarding the digital transformation of a learning space (see Sections 3.2 and 3.3). The implementations might be regarded as situated learning experiences in specific challenging real-world scenarios rather than reports on past or completed situations. This idea represents one of the main contributions of this work: defining implementations as application cases of experiential learning.

The application cases are single exploratory case studies linked to a little-known and unique situation, location, group of people, or event, aiming to explain and gain insights into its particularities, rather than other cases or generic issues [64]. This is a type of qualitative research method that requires the participation and involvement of the researcher and focuses on the contextual characteristics of a study situation, which makes it difficult to generalize the conclusions. Consequently, it can also be claimed that exploratory case studies do not require a hypothesis or research question. The case study illustrates a learning experience in an undergraduate program using an in-depth exploration of the digital transformation of an experiential learning space. A case study method was selected here, as it can be applied to explain the implementation of new methods or techniques where there is only one or a small number of situations or instances. Therefore, no comparisons with control groups were conducted to test hypotheses or develop inferences and further generalizations [65].

### 2.3.3. Stage 3

Finally, the third stage involves the statistical analysis of the collected data (i.e., observation variables) regarding a learning experience to report on these in relation to an application case study (see the results and findings in Sections 3 and 4).

Descriptive statistics, correlation analysis, and Levene's test can help to identify the most significant findings of the collected data regarding a learning experience. The data analysis can provide results related to student learning outcomes, the development of the intended competencies, and students' perceptions of the learning experience. These analyses can only inform us of results related to the corresponding learning experience and do not make possible any generalization to other instances because of the focus of the data on the particular study situation.

Moreover, the results can be assessed according to the criteria of reliability, transferability, and validity [65–67]. Reliability refers to whether or not the collected observations are repeatable or consistently attributed to instances of the same unit of analysis (i.e., learning spaces). Transferability tells us whether (other) researchers can identify new occurrences of the object and where they can consistently use the observations without modification, achieving observational closure. Finally, validity raises the question of how confident one can be in the interpretation of the observations and whether they consistently refer to the same object in the world or reality. These criteria can guide our discussions about the results of this work.

## 3. Results

To exemplify the method presented in Section 2.2, we devised two instances of DeELS for Engineering Education 4.0. For this purpose, this work proposed using the Lean Thinking Learning Space (LTLS) [55], a 2019 QS Reimagine Education award-winning initiative of Tecnologico de Monterrey university in Mexico, to carry out learning experiences for

digital transformation in the form of application cases. Therefore, this research is about transforming the LTLS into a *digitally enabled LTLS*.

The LTLS is an experiential learning space in which industrial engineering students learn lean manufacturing by creating challenging learning experiences, leading to process and operation improvements. However, in the case of this work, this also involves the possibility of identifying the experiential learning activities necessary to carry out digital transformation. That is, the original version of the LTLS only allowed for the undertaking of lean manufacturing learning experiences, but there is now an opportunity to provide engineering students with learning challenges in order to incorporate digital technologies into the learning space and, later, to use this as a digitally enabled LTLS for Education 4.0.

The LTLS is an experiential learning space in which students transform a "*push into a pull*" one-piece flow production process. Learning takes place by transforming a manufacturing process from a push-batch production system into a continuous one-piece flow and, later, into a pull-and-leveled, just-in-time production system [55,62,63].

The production process manufactures wood products through nine operations: circular saw cutting, drawing, belt-saw cutting, grinding, painting, drilling, wheel and shaft drilling, assembly, and inspection and shipping. Students play different roles to satisfy the specific product demands regarding quality, cost, time, safety, and sustainability. The roles comprise operators, supervisors, quality engineers, material handlers, quality inspectors, shipping dispatchers, and plant managers. The whole process is (re-)configured according to specific production needs and learning objectives, resulting in different production system arrangements.

The LTLS involves the deployment of an educational plan according to the specific course requirements, in which the learning objectives, the definition of competencies, learning strategies, learning spaces, challenging learning experiences, and activities are intentionally and coherently defined.

Learning within the LTLS encourages students to create production runs in which problem solving occurs to improve operations such as process stability, variability reduction, quality acceptance, and service level. Hence, the learning experience requires individual and collective problem solving and decision making in order to meet the progressively more astringent (simulated) market and operational requirements.

The original version of the LTLS considers four learning phases required to progress the students' learning of lean manufacturing skills:

- Phase 1 considers that students should familiarize themselves with the whole learning space as a production system, involving the production process, material flows, operations, roles, materials, safety procedures, and the use of tools. Students spend at least three to five hours engaged in training and immersion activities to gain an understanding of how to perform their specific tasks.
- Phase 2 involves the students' achievement of process stability through a batch production plan by implementing standard work, continuous improvement, quality control, the A3 method, process mapping, and essential lean tools, such as the 5 Ss and visual management. It takes approximately fifteen hours to complete this phase.
- Phase 3 has the aim of achieving a one-piece continuous flow in the production process by implementing additional lean tools, such as waste elimination, poka-yokes, quick changeovers, total productive maintenance, and others. It takes approximately another eighteen hours to produce the change in the production process.
- Phase 4 has the aim of moving from a one-piece flow to a pull production flow by implementing kanban, first-input–first-output (FIFO) flows, buffers, and safety stocks, to name a few examples. This stage takes around eighteen hours to complete.

The execution of the learning experience refers to changes in the frequency of the demand and variations in the sequence and combination of manufactured products, which are addressed through specific designs, process innovations, and improvements to meet customer needs. Throughout the four phases, the students continuously execute the experiential learning cycle by (i) perceiving concrete experiences of operations and process

results in regard to their production results versus the targets in order to identify problems, (ii) reflecting upon problems indicated by the results by analyzing and diagnosing the root causes, (iii) designing countermeasures to overcome the problems, and (iv) implementing and monitoring countermeasures to improve the process performance.

### 3.1. Applying the Method to Develop DeELS

3.1.1. Step 1: Define the Educational Purpose

The digital transformation of the digitally enabled LTLS considers the digital competencies of engineering students according to new professional and employment requirements of manufacturing and service companies in diverse sectors and industries. Accordingly, students require digital literacy and the ability to participate in digital transformation projects or functional implementations.

This situation creates the need to develop initiatives aiming to allow students and their education to remain relevant to employers and the labor market. Therefore, by transforming the original LTLS, novel experiential learning challenges can be offered to students that align the digital transformation of the learning space with the educational and industrial requirements. The digital transformation of the LTLS offers learning experiences that present scenarios for solving real-world problems. This alternative allows students from different engineering disciplines to participate in the construction of the digitally enabled LTLS and put their complementary knowledge and abilities into practice within a production process context. Moreover, by creating the digitally enabled LTLS, engineering students can also further recreate novel lean manufacturing learning experiences in a digitally enhanced production environment to improve the process performance and production results. Hence, students are challenged through their learning experiences to select, incorporate, and/or use digital technologies in order to enhance the performance of their operations and production process.

The intervention of the LTLS involves a digital transformation aligned with the educational purpose and the pedagogy behind this learning space. This digital transformation allows for the incorporation of different digital technologies to create a new version of the learning space, as indicated in Figure 3.

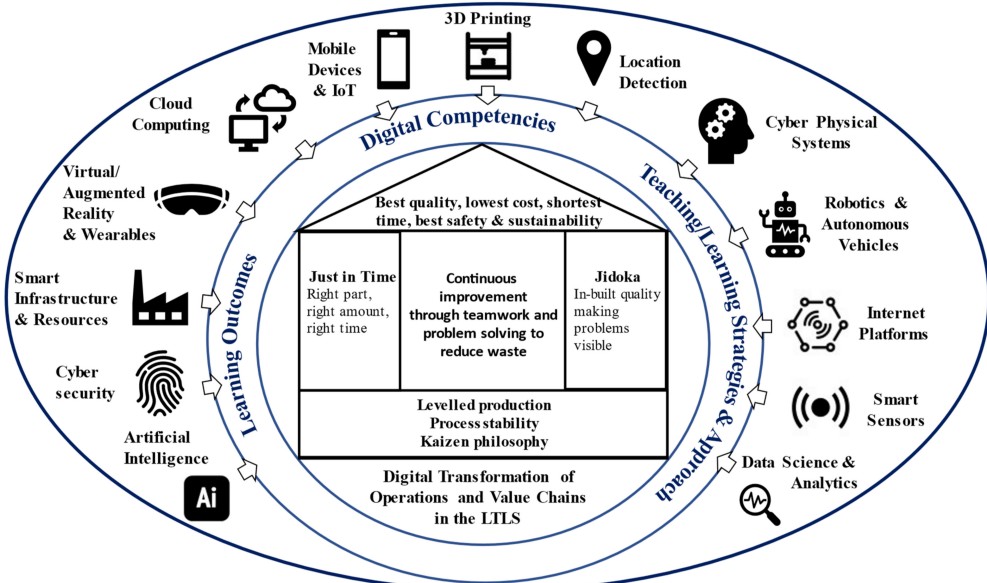

**Figure 3.** Digital transformation technologies for the LTLS (own elaboration) [5]. Requirements of digital technologies translated into the educational components that define lean manufacturing challenging learning experiences within a digitally enabled LTLS.

Figure 3 depicts the Toyota production system framework [68–70], which is at the heart of the core disciplinary concepts and tools used to boost the adoption of digital technologies [56]. Next, a structure of educational components connects the disciplinary knowledge and the digital technologies based on the academic requirements that underpin the students' participation in the experiential learning space [53].

Accordingly, the digital transformation of the LTLS adds value, impacting quality acceptance and on-time deliveries by incorporating digital technologies into the strategies used to improve the drivers of the value chains, processes, and operations. Specifically, these enhanced technologies are intelligent sensors, the industrial internet of things (IIoT), smart cameras, mobile gadgets, augmented reality, collaborative robots, internet platforms, cloud computing, data analytics, and artificial intelligence.

3.1.2. Step 2: Identify the Necessary Experiential Learning Activities

Thus, a challenge-based learning experience enabling students to undertake the digital transformation of operations within the LTLS aims to: (i) *collect data from the operations* about the cycle and throughput times, materials' withdrawal and consumption, quality acceptance, rework, and scrap materials; (ii) *collect data on the operators' tasks*; and (iii) *organize and analyze data* for the purpose of problem solving regarding the incorporation of digital technologies into the production process for performance improvement.

Table 3 summarizes the educational components of the digital transformation of the LTLS, leading to learning activities, according to the framework presented in Figure 1 [10]. This description includes a declaration of purposeful education and experiential learning activities supporting the study and transformation of a production system (i.e., operations and value chains). Students can reflect upon the production system results, conceptualize the design and development of a digital prototype, and experiment to determine the impact of the transformation on the delivery value.

**Table 3.** Description of the digital transformation framework of the LTLS (own elaboration).

| Digital Transformation in Experiential Learning Spaces | Operations and Value Chains |
|---|---|
| Digital Technologies | Smart sensors, IIoT, internet platforms, data analytics, and cloud computing to conform with cyber-physical systems. |
| Learning Outcomes | Digital transformation of processes and operations within the LTLS, working in multidisciplinary teams using digital technologies to create an impact on the operational results in terms of quality, cost, time, and safety. |
| Competency Development | The ability to evaluate various technologies with openness in order to search for and implement relevant alternatives for the transformation of professional practice, according to economic, environmental, social, political, ethical, safety, hygiene, and manufacturability restrictions. |
| Learning Strategies | Competency-based education; challenge-based learning; experiential learning. |
| Learning Space | Lean thinking learning space (LTLS). |
| Challenging Learning Experience | Digital kanban system; smart double-bin kanban. |
| Experiential Learning Activities | (i) Study the production system. (ii) Analysis of the production system results. (iii) Design and develop digital prototypes to enhance operations. (iv) Prototype implementation, experimentation, and validation. |

The methodology of this work helps one to plan and observe the students' performance and participation in the learning experiences. In this way, the study of the observation variables defined in Section 2.3.1 involves the data collection and analysis incorporated into the methodology.

### 3.1.3. Step 3: Conceptualize a Purposeful DeELS

The LTLS allows students to undertake production plans and develop future creations by digitally transforming and upgrading the original learning space, as presented in Figure 4. In this case, Figure 4 shows a proposed version of a digitally enabled LTLS involving (i) a pull production process; (ii) nine sequential production operations; (iii) operators and supporting roles that perform operations; (iv) lean manufacturing resources embedded in the production process, such as kanbans, a heinjunka box, standard work and job instructions, and andon posts; (v) lean manufacturing methods and tools used for problem solving, such as Kaizen and the A3, aiming to improve operational efficiency; (vi) diverse digital technologies such as collaborative robots and autonomous vehicles, aiming to improve cycle times, and gadgets, sensors, and mobile displays aiming to enhance process measurements, monitoring, inspection, and control; and (vii) data analytics, an internet network, and computing technologies, aiming to process data and support decision making. Examples of possible learning experiences include the incorporation of digital kanbans for material supply and replenishment and robotics and autonomous vehicles to improve the production cycle times, service levels, and inspection cameras for quality acceptance. Other alternatives can refer to wearable devices, aiming to collect the operators' biometric signals in order to correlate them with process performance and the use of virtual or augmented reality for job instructions and standard work, as well as smart cameras and sensors for quality inspection, among other tasks. These possibilities define a research agenda aiming to explore diverse, challenging learning experiences within the LTLS that improve its operation and support the development of disciplinary and digital competencies. Hence, this work reports on the first effort to digitally transform the LTLS (see Section 3).

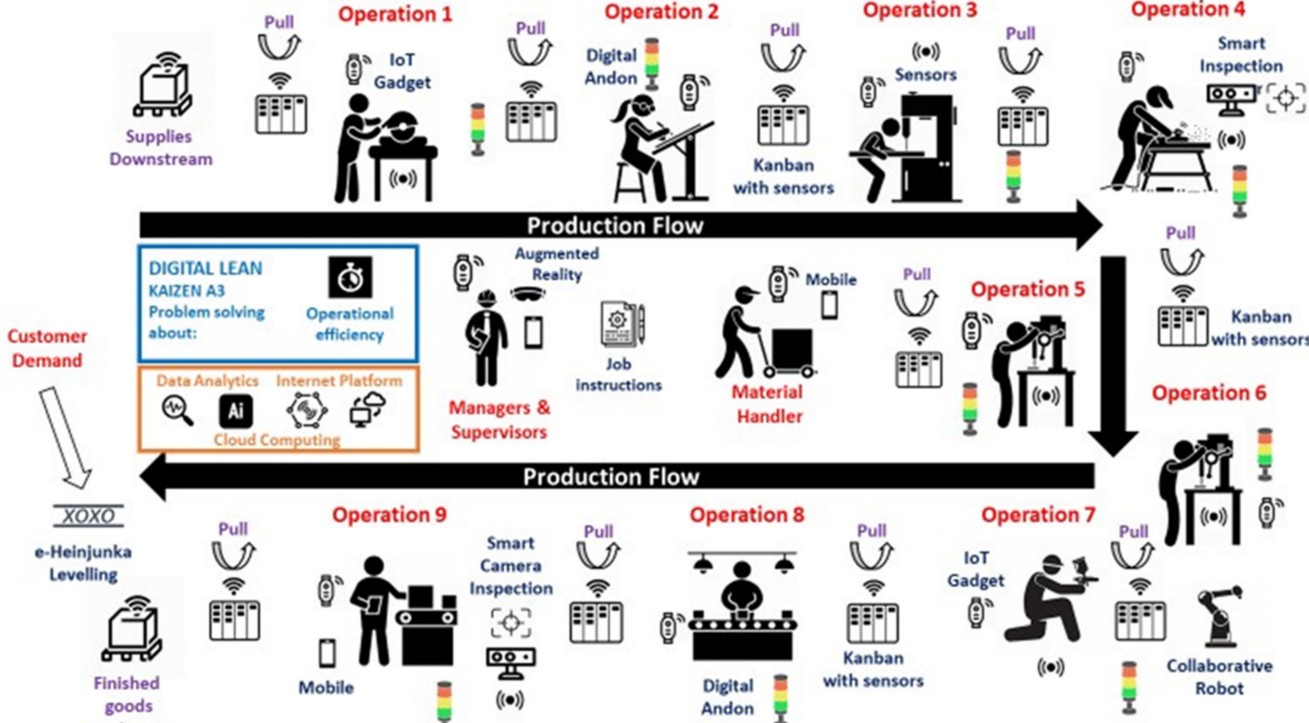

**Figure 4.** The digital transformation of the LTLS (own elaboration) [10]. A proposed integration of digital technologies into operations to enhance the process performance and production results in a lean manufacturing experiential learning space.

3.1.4. Step 4 Design Challenging Learning Experiences

Sections 3.2 and 3.3 describe two instances of challenging learning experiences that resulted in a digital kanban system and a smart double-bin kanban. These experiences were designed to enhance process flows and material resupply/replenishment using intelligent sensors connected to the IIoT, wearables, data analytics, and cloud computing in order to monitor process times. This transformation also included dashboards to track work-in-process materials. Each application case considered the student learning outcomes and competencies of the selected courses. Hence, these two learning experiences represent the first step in the digital transformation of the LTLS. The two instances involved engineering students from different disciplines, not only industrial engineering, which required a debrief of the production systems and lean manufacturing concepts to enable the students to understand the relevance of implementing digital technologies to enhance process performance.

*3.2. The Digital Kanban System Application Case*

In the first digital transformation learning experience within the LTLS at the Tecnologico de Monterrey Campus Puebla, industrial engineering students undertook the challenge of developing a novel proposal of a digital kanban system using Industry 4.0 technologies in order to simplify the replenishment and availability of materials [71–73].

During the last three years, the engineering courses conducted in the LTLS have focused on increasing the engineering- and sustainability-related competencies of the students. However, the automotive industry cluster, located in the area of the LTLS in Mexico, introduced academic instructors to the existing limitations on the production capacity in the industry in order to satisfy the current in-time demands of mass customization. Other issues concerning shortages of materials and deficiencies in supply chain information flows pointed to the urgent need to improve operational efficiency and eliminate waste. For this reason, the design of the digital kanban system, as a challenging learning experience using Kolb's experiential learning cycle, incorporated Industry 4.0 enablers and components, so that the students could develop digital transformation competencies to meet industrial requirements.

This learning experience challenge was conducted in the fall semester of 2018 as part of the Operations, Design, and Optimization Laboratory course (IN3038), with fourteen students of industrial engineering, two students of information technology engineering, one student of robotics, and two students of mechatronics engineering. The students worked in a multidisciplinary group, supporting each other, contributing according to their study disciplines, and complementing their engineering knowledge and skills through role playing within the learning experience. This course learning objective points to the students' ability to select an appropriate set of tools with which to analyze and enhance a production system's performance. The student learning outcome is to apply the acquired knowledge in order to design, analyze, and improve a production process through a laboratory experience.

With the feedback of one automotive company, which explained to the students the relevance of this type of device in real-world production processes, the instructors and students worked to create a digital kanban system as a hyper-connected device using sensors, actuators, and other devices for the interactive, interconnected, inter-cooperative, and inter-functional coordination of the operations [73–75]. The expected result was to develop an integrated network in a smart factory or intelligent process. The anticipated outcome was that the experience would contribute to the creation of product value regarding product availability, service level, and digital integration [76].

Figure 5 shows the digital kanban system prototype, including a smart bin using an IIoT component interconnected with a cyber-physical system through an internet network. This system provided the operators and supervisors with real-time replenishment information about the product supply to the workstations through software applications and information technology through wearable devices, such as smartphones or smartwatches.

The operators and supervisors received data analytic information as key performance indicators and process variables on dashboards with a graphical user interface. Additionally, the industrial engineering students were provided with basic notions of cyber-physical systems and off-the-shelf solutions for the development of the digital device. Therefore, the digital kanban bin followed the cyber-physical system specifications, providing a plastic container with incoming and outgoing material areas. The kanbans had sensors with an infrared (IR) transmitter and emitter–receiver pairs mounted on plastic bins to detect the operator's actions and material flows. Each pair drew a cross-sectional infrared transmitter line that was interrupted when it detected something in the corresponding area (see Figure 6).

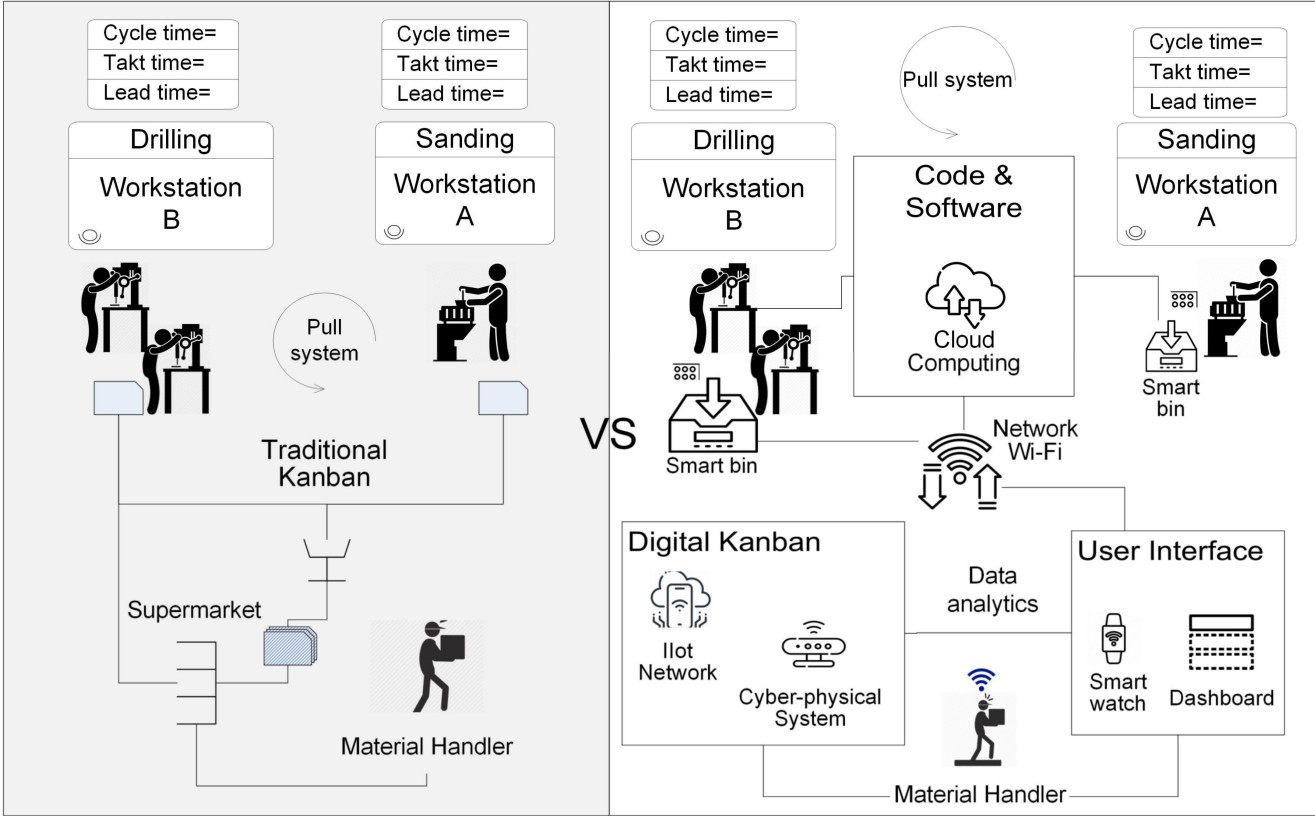

**Figure 5.** Traditional kanban system vs. digital kanban system (own elaboration). Kanban inventories of the workstations are pulled by downstream operations and replenished by upstream operations using (i) production and transportation cards in traditional kanban systems and (ii) electronic signals of smart watches, phone apps, and dashboards in the digital kanban system. The transformation of the traditional kanban system into a digital kanban system consists of a learning challenge within the LTLS.

The dynamic process of interaction between the cyber-physical systems, IIoT, wearables, and the software platform is visualized in Figure 7 as a pull system restocking loop.

All the engineering students and six faculty members participated in the learning experience to design and produce a product with a social impact [63]. The students defined the initial configuration of the production process, starting with a push system that progressively transformed into a pull system. Subsequently, the students incorporated the digital kanban system into the workstations, overcoming the digital transformation problems of replenishment and the cycle times. Finally, the students elaborated and presented a technical portfolio providing evidence of the development of their digital competencies during the learning experience.

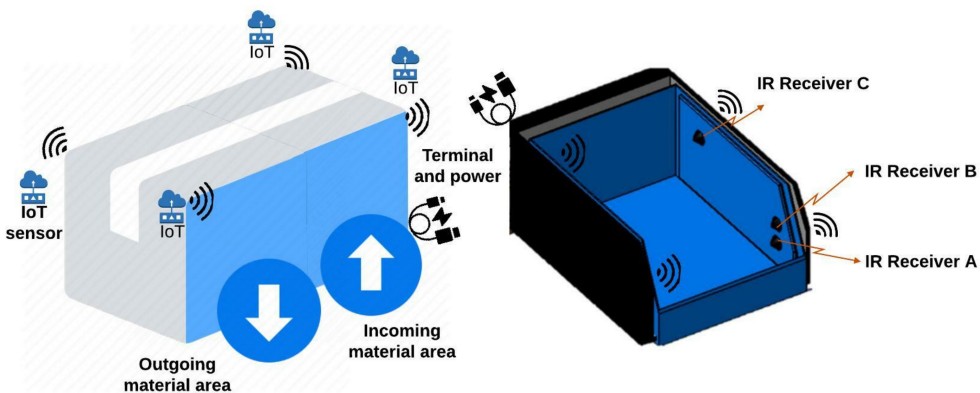

**Figure 6.** Digital kanban bin as an IIoT component (own elaboration).

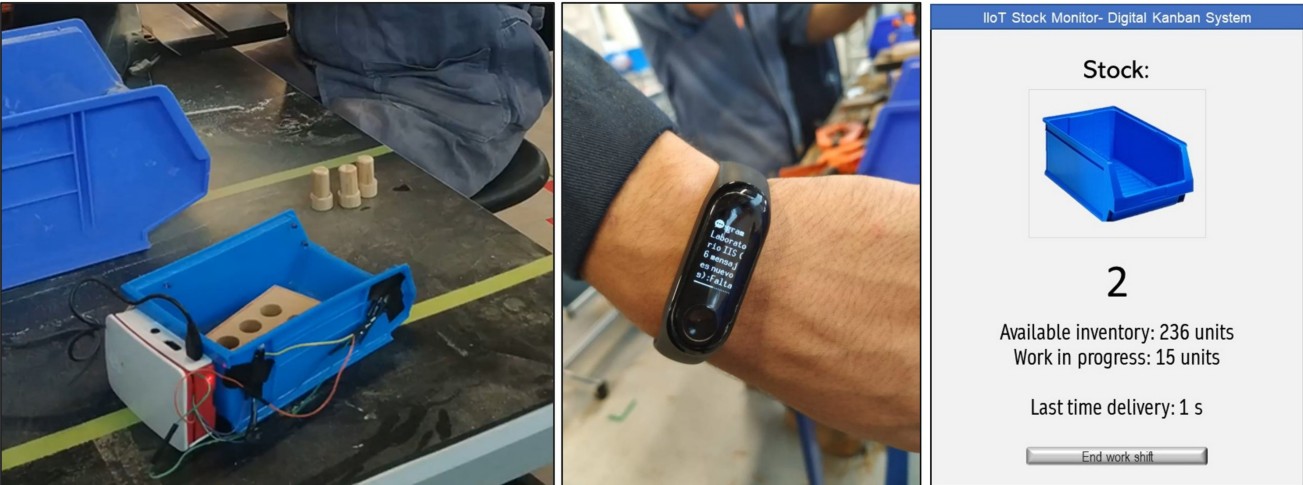

**Figure 7.** User interfaces: wearables and the front end of the digital kanban system. IIoT signal of a smart watch (left) and digital kanban system dashboard and interface (right).

### 3.3. The Smart Double-Bin Kanban Application Case

The second case, the smart double-bin kanban, was developed by undergraduate students in a challenging learning experience based on Industry 4.0 technologies, aiming to improve inventory management in a production process. This project involved the design of an intelligent double-container system for mechatronics engineering students undertaking the laboratory courses of Industrial Networks (MR2019), Mechatronic Instrumentation (MR2005), and Logic Automation (MR2002). The container system provided a technological enabler to control inventory replenishment using visual, automatic, and real-time information. In this case, the mechatronics engineering students were tasked with working in the LTLS to provide them with an experiential learning space with a focus on an industrial production process.

This system consisted of two bin boxes with weight sensors and integrated traffic lights to inform the operators about the material flows (with a green light for removing materials, red light for not removing materials, and amber light for replenishing materials). These containers were connected to an interface that monitored the inventory changes in real time and a material replenishment alert launched to notify the user of product shortages (see Figure 8).

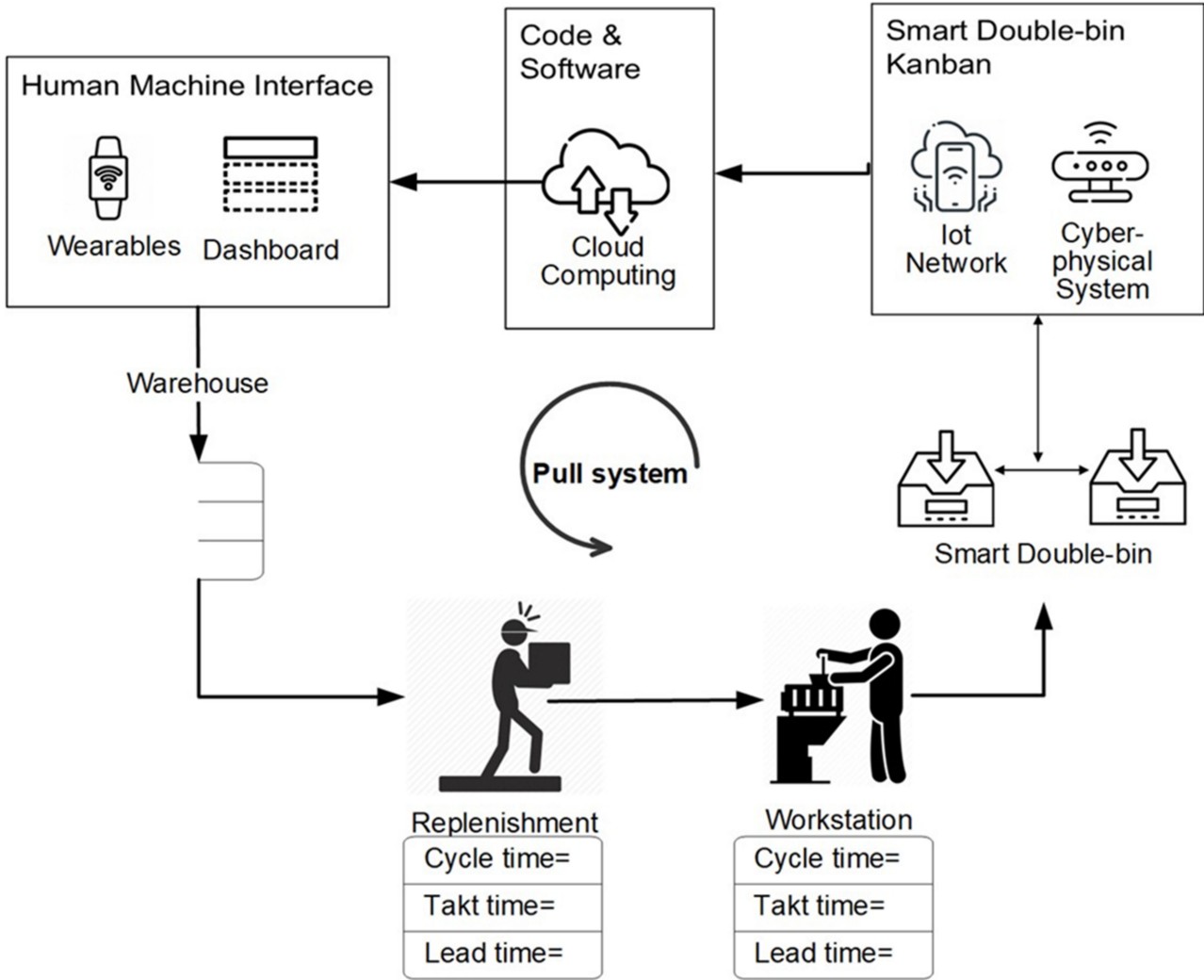

**Figure 8.** Replenishment system of the smart double-bin kanban system (own elaboration). A learning challenge is carried out in the LTLS when a card kanban system is transformed into a smart double-bin kanban system using the automation technologies of sensors, micro-controllers, light signals, and dashboard displays to control replenishments in the operations.

The learning objectives of these courses point to the students' ability to evaluate, configure, and apply different types of industrial communication networks in order to solve automation and industrial informatic problems. The student learning outcome of this course is to configure network devices, field networks, and industrial ethernet networks using various programmable logic controllers. The students are to identify and solve device interconnectivity problems throughout industrial communication networks, illustrate processes through human–machine interfaces, and implement broad-application industrial networks.

In the study, this challenge of the learning experience aimed to evaluate the digital transformation competencies of the students by identifying production and technical requirements, evaluating technological alternatives, designing a viable solution, and creating a functional, double-container kanban system prototype with 4.0 technologies.

In this learning challenge, the faculty instructors combined industrial engineering with mechatronics engineering concepts to enable the students to develop a digital solution aiming to replenish production lines or inventory materials in storage. Figure 8 shows the conceptualization of a digital kanban system aiming to guide the students' design efforts in their learning space. At the end of the study course, the students presented their

prototypes or simulations of technological enablers to a panel of faculty members from the Mechatronics and Industrial Departments. They evaluated the prototype results according to the disciplinary course rubrics.

The challenging learning experience was carried out for the first time in 2019 with twelve mechatronics students in the fall semester. These students, enrolled in the Industrial Networks Laboratory course, formed three teams of four students. Each team had to design three prototypes using programmable logic controllers (known as PLCs), as shown in Figure 9.

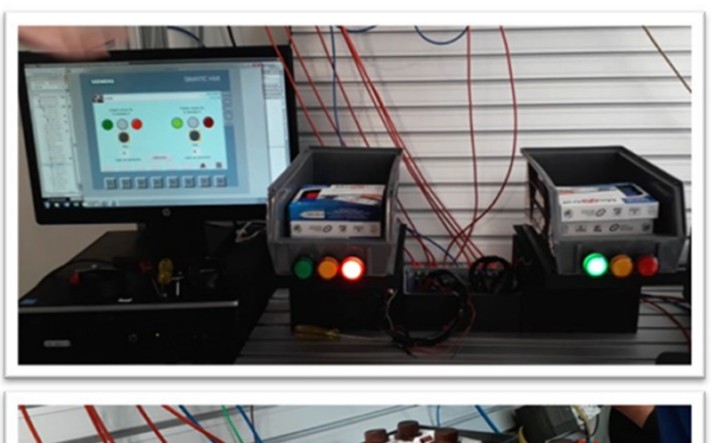
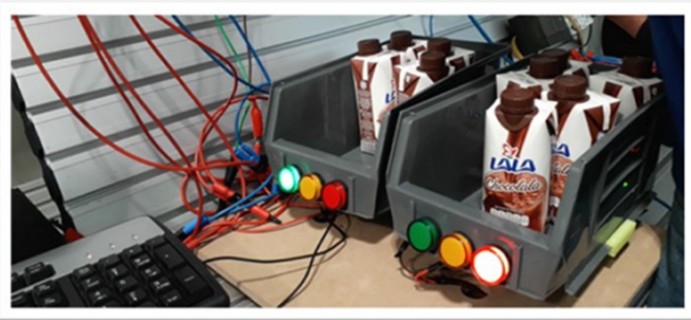
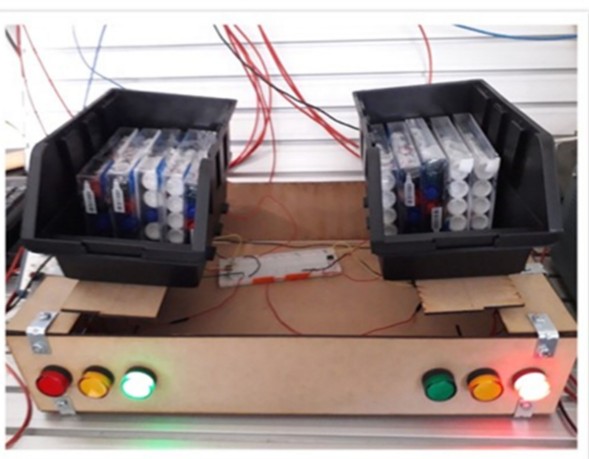

**Figure 9.** Three prototypes of the smart double-bin kanbans were developed in the first learning experience.

The prototypes were developed using CAD design, instrumentation, programming, communication systems, and cloud computing. Each prototype still had opportunities for improvement but allowed the students to undertake a relevant learning experience.

Later, the challenging learning experience was carried out again in the spring semester of 2020 as part of the Mechatronics Instrumentation and Logic Automation laboratory courses. Students had to improve the smart kanban without using PLC. However, because of the COVID-19 contingency, the students changed their development design to simulated prototypes using the Thinker Cad design tool, with ultrasonic sensors to monitor the percentage of the products refilled in the containers and the automatic replenishment system. As for the smart kanban, presence sensors inside the two containers captured the amount of product in real time. A microcontroller adequately activated the three colored light indicators (green, amber, and red). Regarding the replenishment of the containers, the students performed a successful simulation using the microcontroller (see Figure 10). The software programming was run in Arduino, with its operating logic showing satisfactory results.

A total of twenty-nine students and four faculty members participated in the learning experience to design a prototype of the smart double-bin kanban. The students defined the initial configuration of the smart double-bin kanban system, starting with their manual process knowledge. Subsequently, the students incorporated the automated smart double-bin kanban into the warehouse shelves, overcoming the digital transformation problems of replenishment and the cycle times. Finally, the students elaborated and presented a

prototype providing evidence of the development of their digital competencies during the learning experience.

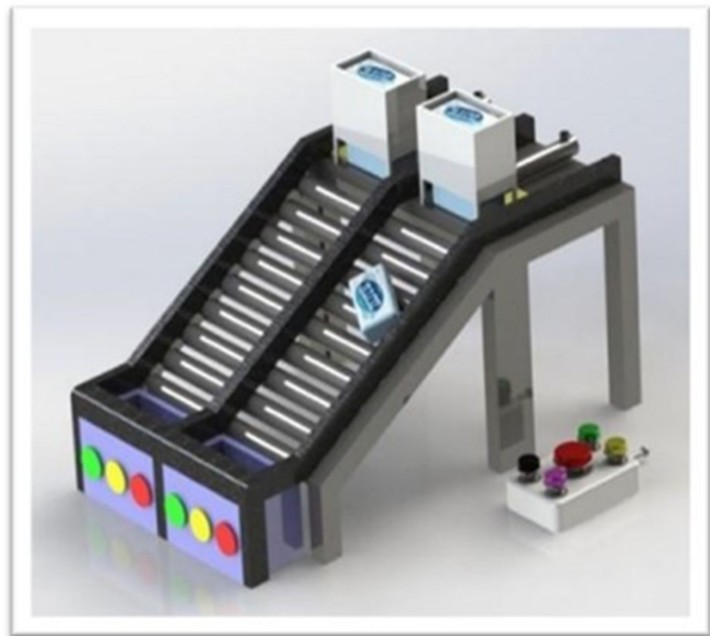

**Figure 10.** Prototype of the smart double-bin kanban and an automatic replenishment system developed in the second learning experience using Thinker Cad during the COVID-19 pandemic.

### 3.4. Qualitative and Quantitative Results of the Learning Experiences

The following sections describe the qualitative and quantitative results and analysis of the challenging learning experiences and their implementations according to the methodology described in Section 2. Table 4 summarizes information about the two application cases, campus location, course names, and populations of students.

**Table 4.** Summary of the learning experiences as application cases.

| Application Cases of Learning Experiences | The Geographic Location of the Learning Space | Course ID and Name | Student Population |
|---|---|---|---|
| Smart double-Bin Kanban | Mexico State Campus | MR2019 Industrial Networks | 14 |
| | | MR2002 Logic Automation | 12 |
| | | MR2005 Mechatronic Instrumentation | 3 |
| Digital Kanban System | Puebla Campus | IN3038 Operations Design and Optimization Laboratory | 14 |

#### 3.4.1. Competency Assessment

Appendix A shows the assessment rubric with three levels of evaluation of the students' competency achievements. Level A is the highest score value. This categorical numerical variable uses a Likert scale ranging from 1 to 3, where level A = 3 [77]. These three levels of achievement correspond to the assessment criteria defined by the Tec21 Educational Model of Tecnologico de Monterrey [29]. Data were collected from each student for the evaluation of their competencies.

Table 5 details the descriptive analysis of the students' assessments and digital competency achievement levels. Accordingly, all the courses presented similar results with regard to the DeELS. Although the Logical Automatisms course results are below the interme-

diate competency level B, the standard deviation (StDev) is similar to those of the other courses. In contrast, the median and the mode results are "2", which corresponds to level "B" of the competency achieved by the students. The findings, in both cases, show that the students developed the same competency level through the two different courses. This is a consideration for future instances of DeELS, as further evidence should be collected in other application cases.

**Table 5.** Descriptive statistical analysis of the competency achievement level.

| Application Case for Digital Transformation | Course ID | Course Name | N | Mean | Assessment Level | StDev | Median | Range | Mode | N for Mode |
|---|---|---|---|---|---|---|---|---|---|---|
| Smart Double-Bin Kanban | MR2019 | Industrial Networks | 14 | 2.29 | B | 0.83 | 2.5 | 2 | 3 | 7 |
| | MR2002 | Logical Automatisms | 12 | 1.92 | C | 0.79 | 2 | 2 | 2 | 5 |
| | MR2005 | Mechatronic Instrumentation | 3 | 2 | B | 0 | 2 | 0 | 2 | 3 |
| Digital Kanban System | IN3038 | Operation Design and Optimization Lab | 14 | 2.36 | B | 0.75 | 2.5 | 2 | 3 | 7 |

This research shows similar results in regard to the development of digital transformation competencies through the two challenging learning experiences, independent of the differences in geographical context, type of academic program, courses involved, instructors, and educational partners. Accordingly, the data analysis shows that the results of the application cases support the contribution of the DeELS to the development of digital transformation competencies among students.

In summary, there is an acceptable level of achievement according to the learning objectives in the development of digital transformation competency in the two cases. These results suggest that the *students learned to select and apply Industry 4.0 technologies in manufacturing/production environments.*

The students were also capable of designing prototypes with some acceptable deficiencies in their operation. Moreover, they could incorporate engineering techniques and tools that support the digital transformation of a business (this corresponds to level B in the rubric).

3.4.2. Final Project Quantitative Evaluation

The second analysis of the results corresponded to the quantitative evaluation of the project reports based on the two challenging learning experiences.

Table 6 shows the main findings, highlighting the different standard deviations and means of the grades for each learning experience based on a 0–100 grading scale, where 70 is the passing grade (see Appendix B for the Project Report Evaluation Rubric).

**Table 6.** Students' final project reports grades.

| Learning Challenge | Course Code | N | Mean * | StDev | Median | Range | Mode | N for Mode |
|---|---|---|---|---|---|---|---|---|
| Smart Double-Bin Kanban | MR2019 | 14 | 88.64 | 9.76 | 91.6 | 24.65 | 98.5 | 4 |
| | MR2002 | 12 | 85.05 | 6.91 | 87.48 | 17 | 89.47 | 3 |
| | MR2005 | 3 | 100 | 0 | 100 | 0 | 100 | 3 |
| Digital Kanban System | IN3038 | 14 | 95.43 | 5.17 | 96.5 | 20 | 98 | 4 |

* Response variable. Y = Project evaluation.

The grading of the evaluated reports was high, as indicated in Table 6. The means for the two learning experiences ranged from 85 to 100. The overall grade scale range varies from 0 to 100. Therefore, it is evident that the quantitative evaluation of the documented projects resulted in high values. Furthermore, the mean of the lowest group (MR2002) was 85.05, and the median was 87.48, which confirms that the grade values showed a shift

toward the high values on the scale. Therefore, *the tutors' evaluation of the students' reports on the digital transformation of the two kanban systems was favorable.*

However, Figure 11 shows the confidence intervals of the means. The figure graphically shows that, in terms of the 0–100 score evaluation, the courses were not similar. This analysis shows wide variability in the final project report scores. These differences may reside in the evaluation of the final projects because of the different grading criteria of each instructor, the academic quality of the reports delivered by students, and the different disciplinary and technical knowledge objectives defined for each of the disciplinary courses.

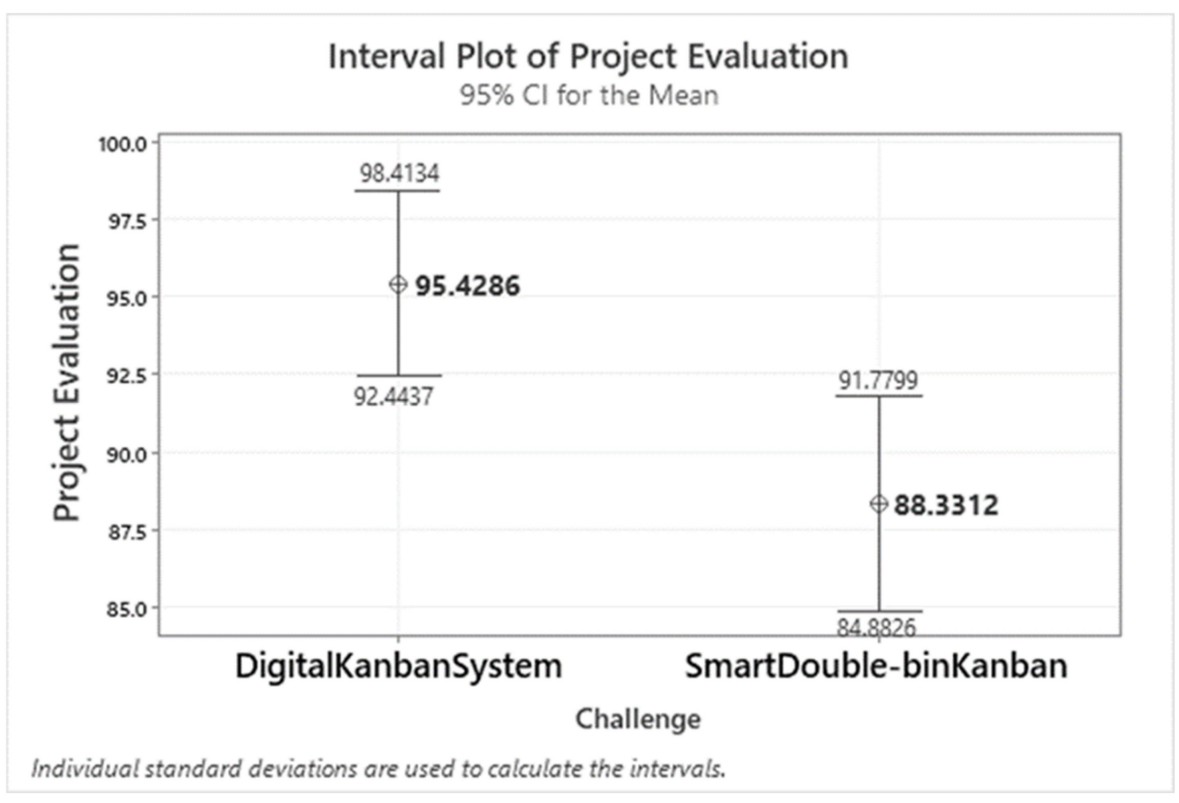

**Figure 11.** Confidence intervals of the means: project report evaluation.

Figure 12 shows the multiple comparisons and Levene's test confidence intervals for the standard deviations (both with a *p*-value of >0.05). This analysis shows the standard deviation of the numerical variable of the final project evaluation for each of the two challenging learning experiences. A significance level of 5% suggests that the population standard deviations of both case studies are statistically equal. In detail, Figure 12 presents greater variability in the confidence interval of the digital kanban system than in that of the smart double-bin kanban project.

Therefore, after the analysis of Section 4.1, we are left with contrasting findings in evaluating the achievement levels of the competencies (A = 3, B = 2, C = 1) and project evaluation grades (0–100), as there are similar differences in the results for the two learning experiences. There is a difference in the numerical achievement level because of the possible dissimilarity between the instructors' evaluation criteria and the use of grading scales when assessing the reports.

Hence, the students' level of competency achievement might not correspond to the grades of their final project evaluations. On the other hand, the numerical evaluations of all the courses had average grades above 85, corresponding to a high achievement level above the level of passing or acceptable results. Thus, the *students achieved good to excellent grades in the technical documentation of their projects in their courses.*

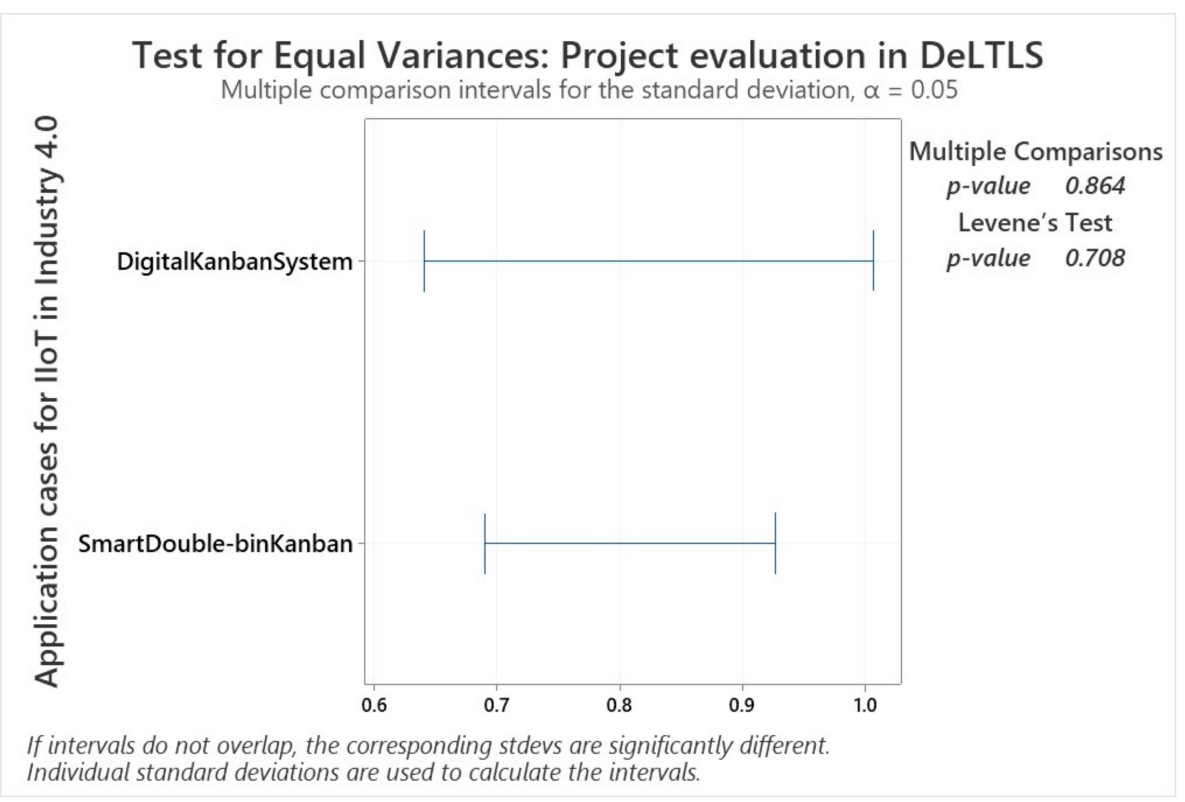

**Figure 12.** Test of the equal variance results: project report evaluation.

### 3.4.3. Students' Opinions

After completing the challenging learning experiences, the students received a survey to provide feedback on four relevant variables representing their perception of their learning. The four observed variables, based on Marzano's self-system achievement, were relevance, motivation, interest, and a self-assessment of competency development, measured by a Likert scale ranging from 1 to 5, where 1 indicates the lowest or worst opinion and 5 represents the highest or best opinion possible (See Appendix C) [77].

It was necessary to explore student perceptions because this was a crucial parameter and a reference point indicating whether the students perceived that they had made progress in their learning and competencies development. The students may have recognized the applicability of the DeELS as a relevant, motivating, and exciting learning experience, in addition to self-diagnosing their achieved level of digital transformation competency. Therefore, their opinions are decisive for this work's continuous improvement and development as an educational innovation.

The collected answers provided in the opinion survey were voluntary and anonymous. The survey answer rate was 14 students out of a total population of 14 students in the case of the digital kanban system learning experience and 12 out of 29 students in the case of the smart double-bin kanban experience.

Tables 7 and 8 show the descriptive analysis, including the means, medians, modes, ranges, interquartile range (IQR), and $N$ for mode. Figures 13 and 14 show the histograms of the students' answer distribution (on a 1-to-5 Likert scale) for each survey question. Moreover, Figures 15 and 16 refer to the boxplot results, describing the data dispersion.

**Table 7.** Students' opinions and descriptive statistics analysis of the digital kanban system.

| Q * | Variable | Mean | StDev | Min | Median | IQR | Mode | N for Mode |
|---|---|---|---|---|---|---|---|---|
| Q1 | Relevance | 4.14 | 0.770 | 3 | 4 | 1.25 | 4 | 6 |
| Q2 | Motivation | 4.50 | 0.519 | 4 | 4.5 | 1.0 | 4, 5 | 7 |
| Q3 | Interest | 4.36 | 0.745 | 3 | 4.5 | 1.0 | 5 | 7 |
| Q4 | Competency development | 4.36 | 0.745 | 3 | 4.5 | 1.0 | 5 | 7 |

* Quartile.

**Table 8.** Students' opinions and descriptive statistics analysis of the smart double-bin kanban.

| Q * | Variable | Mean | StDev | Min | Median | IQR | Mode | N for Mode |
|---|---|---|---|---|---|---|---|---|
| Q1 | Relevance | 4.25 | 0.754 | 3 | 4 | 1.0 | 4, 5 | 5 |
| Q2 | Motivation | 4.25 | 0.754 | 3 | 4 | 1.0 | 4, 5 | 5 |
| Q3 | Interest | 4.33 | 0.778 | 3 | 4.5 | 1.0 | 5 | 6 |
| Q4 | Competency development | 4.25 | 0.622 | 3 | 4 | 1.0 | 4 | 7 |

* Quartile.

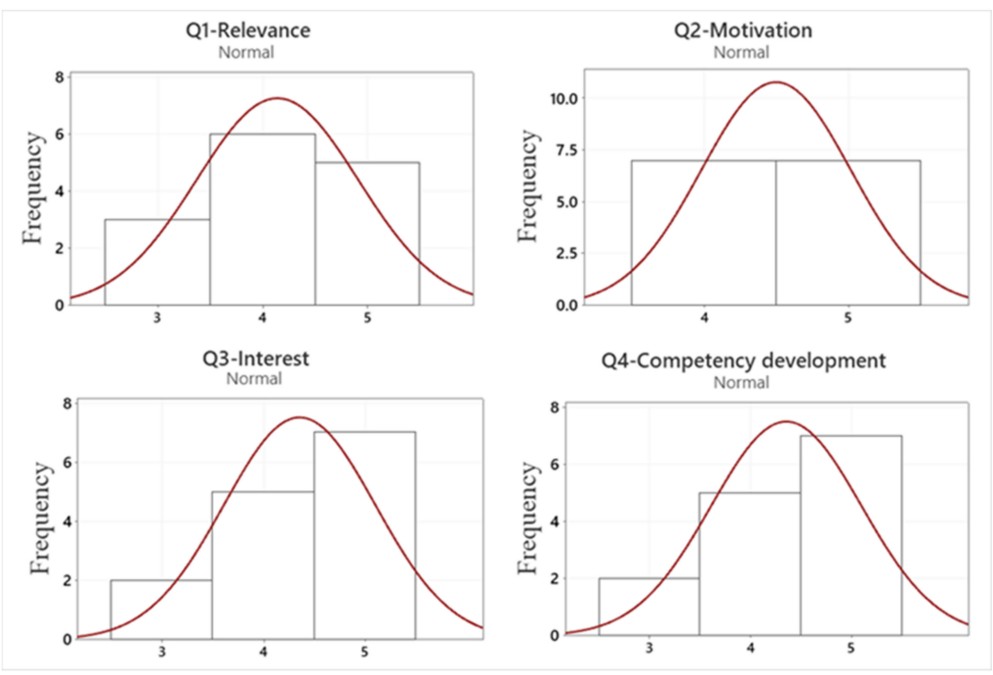

**Figure 13.** Histogram of the students' answers for the *digital kanban system*, *N* = 14 (see Appendix C).

The results indicating the motivation and interest in the digital kanban system learning experience have slightly higher values than in those of the students who experienced the smart double-bin kanban, referring to the mean, median, and mode. In contrast, the relevance was slightly higher in the case of the smart double-bin kanban in regard to the mean. However, the results suggest highly similar results for these variables.

Additionally, the variability in the students' responses, in both cases, was highly similar in regard to the IQR. However, it should be noted that in the case of the digital kanban system, the standard deviation of the motivation variable was the smallest of all the studied factors. Referring to the competency development self-assessment results, the mean, median, and mode were higher than those for the digital kanban system.

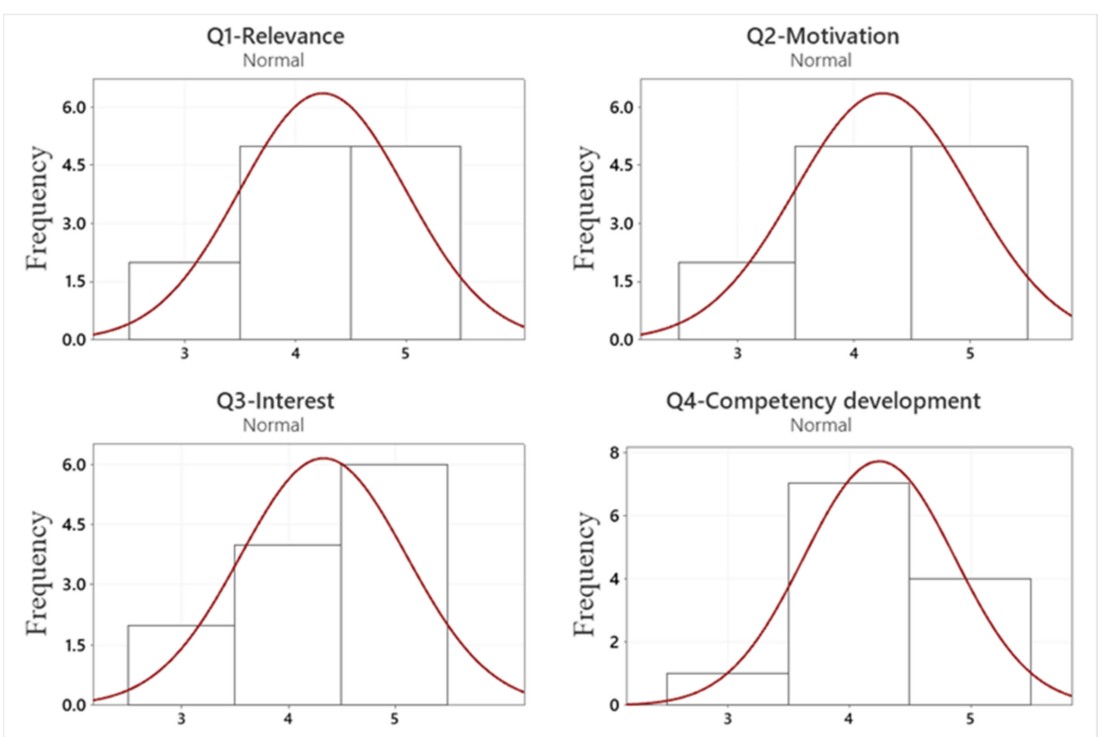

**Figure 14.** Histogram of the students' answers for the *smart double-bin kanban system*, *N* = 12 (see Appendix C).

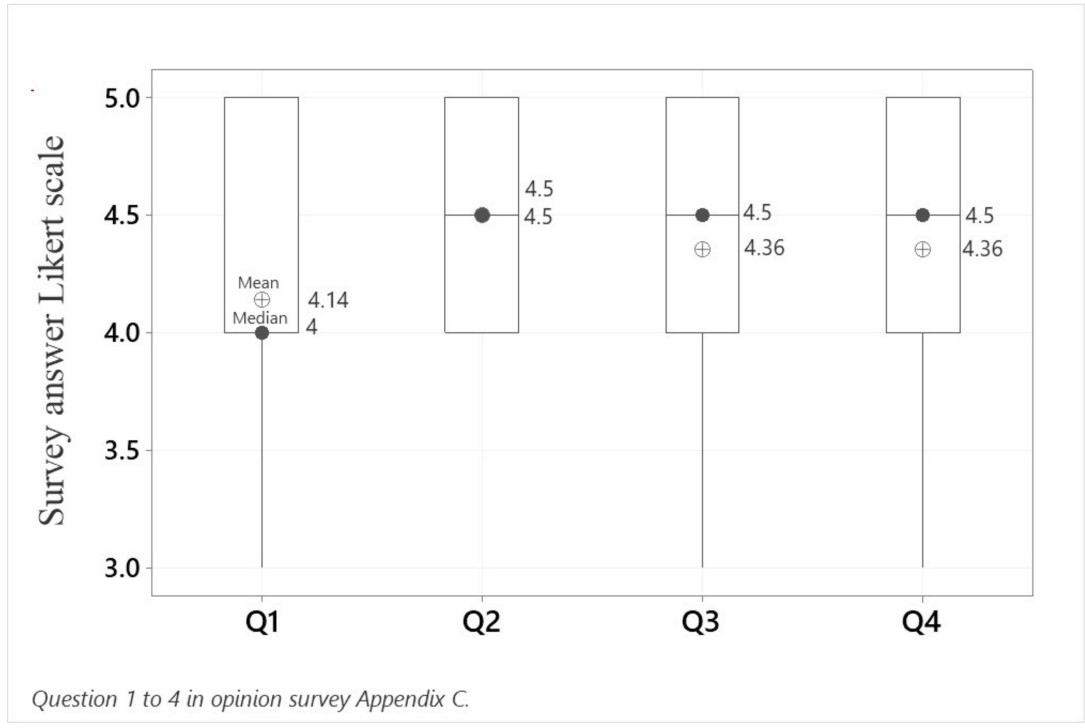

**Figure 15.** Box plot of the students' opinions and answers for the digital kanban system, Q1: Relevance; Q2: Motivation; Q3: Interest; and Q4: Competency development, *N* = 14 (see Appendix C).

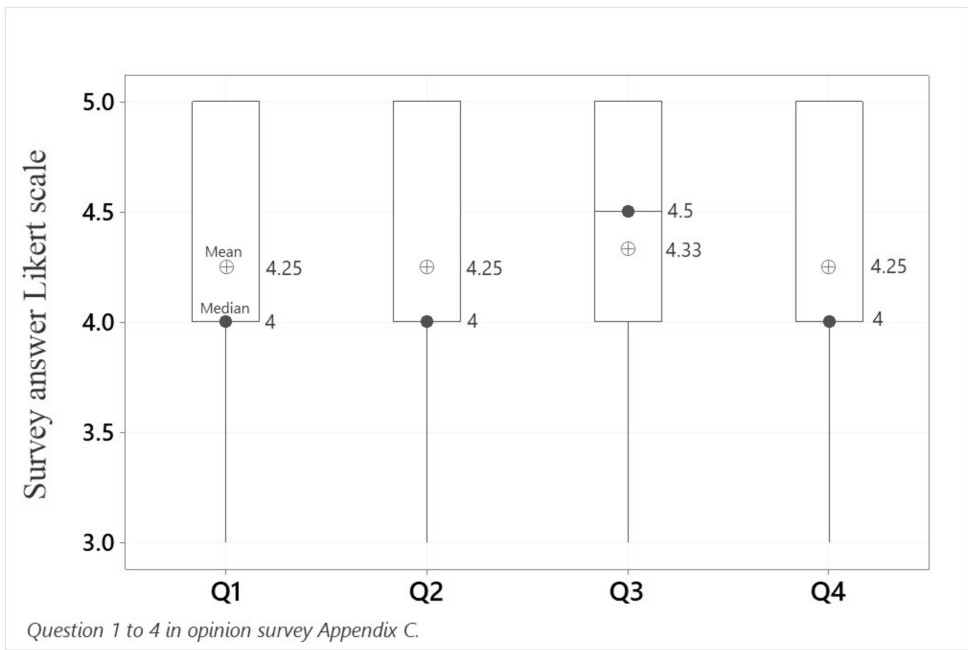

**Figure 16.** Box plot of the students' opinions and answers for the smart double-bin kanban system, Q1: Relevance; Q2: Motivation; Q3: Interest; and Q4: Competency development, *N* = 12 (see Appendix C).

In Figure 15, the boxplot of the digital kanban system shows that in Q3 and Q4, there is a perception of high student interest and competence development. In Figure 16, the boxplot of the smart double-bin kanban indicates that the results of the students' answers to questions Q1, Q2, and Q3 were very similar. The students considered their learning experience as relevant and highly motivating, with a good perception of competence development.

Therefore, most of the students showed a high interest in these learning experiences. They perceived a high level of development in their digital transformation capacity to engineer solutions using Industry 4.0 enablers under the existing restrictions.

Subsequently, a correlation analysis identified dependence between the variables in questions Q1–Q4 to determine their interrelation, interaction, or interdependence based on the student perception survey instrument. If any of the question variables are correlated, the results suggest that the DeELS contributes to the development of relevance, interest, or motivation among learners, thus directly developing their digital transformation competency (See Appendix A). This also points to other variables that should be studied in the future. Table 9 presents the Pearson correlation analysis results, with a moderate correlation between the qualitative studied variables of Q1–Q2 and Q2–Q3, based on the following criteria: 0.00–0.19 very weak; 0.20–0.39 weak; 0.40–0.59 moderate; 0.60–0.79 strong; and 0.80–1.00 very strong [78–81].

Table 9 shows results indicating a moderate correlation between the variables studied in the student opinion survey Q2 and Q1, as well as Q3 and Q2, and a weak correlation between Q3 and Q1, Q4 and Q2, and Q4 and Q3. These results suggest that *the higher the relevance of a challenging learning experience within a DeELS is, the greater the motivation, interest, and self-assessment regarding the development of the competencies of the students will be*.

**Table 9.** Pearson correlation analysis between the observed variables (questions in Appendix C).

| Question | Q1—Relevance | Q2—Motivation | Q3—Interest |
|---|---|---|---|
| Q2—Motivation | **0.423** | | |
| Q3—Interest | 0.222 | **0.452** | |
| Q4—Competency development | 0.193 | **0.331** | **0.373** |

## 4. Discussion

### 4.1. Learning Experience Findings

From the analysis of the results, this work concludes that the application cases presented appropriate instances of Education 4.0 based on the students' achievements in terms of their grades, competency assessments, and opinions. The DeLTLS can help to develop student learning outcomes in lean manufacturing for the purpose of process improvement and digital competency in Education 4.0.

To sum up, the students were able to successfully engineer a solution based on a cyber-physical kanban system to replenish materials within the production process and meet the operational performance requirements. On the one hand, the industrial engineering students were able to collaborate with the engineering students of other disciplines to incorporate digital technologies into their operations. Moreover, the engineering students from other disciplines were able to understand and develop a functional digital solution in an industrial context.

The two application cases helped us to explore the students' learning achievements. The descriptive statistical analysis showed that the two challenging learning experiences produced similar contributions in terms of the acceptable achievement levels in the development of digital transformation competency. Moreover, there were contrasting findings based on the competency evaluation and report grades for the two learning experiences, as there were similar differences in the results. These differences in the students' numerical grades might have arisen because of the possible dissimilarities between the evaluation criteria or the quality of the learning outcomes, such as reports and exams. Hence, the students' level of competency achievement might not have correlated with their grades in their final project evaluations or final grades.

Moreover, the results of the statistical analysis suggested that the students considered their participation as highly interesting, very motivating, and relevant. The correlation analysis showed that the higher the relevance of the challenging learning experience within a DeELS, the greater the motivation, interest, and self-assessment of the students in regard to their competency development. This correlation is crucial for the design of new learning experiences and the management of students' engagement during their learning activities. Thus, relevance, student interest, and motivation are paramount for developing competencies.

In this sense, referring back to the expected achievements of Education 4.0 for engineering education, we could argue that the DeELS contributed to the development of digital competency among the students regarding (i) the evaluation of digital technologies to select and implement relevant alternatives, (ii) the transformation of a production process as an experiential learning space in a professional practice scenario, and (iii) the consideration of the manufacturing restrictions, economic limitations, academic ethical considerations, and safety and hygiene procedures required to achieve their learning outcomes. The DeELS created a dynamic in which the students improved their innovation, creativity, and interpersonal skills in Education 4.0. Moreover, the learning spaces provided the students with a personalized, problem-based, collaborative, and student-driven learning experience.

Throughout this work, multidisciplinary work was necessary for the students in order to develop comprehensive and robust engineering solutions for the digital transformation of the LTLS. The challenging learning experiences provided the opportunity to have several engineering disciplines included in the learning experiences. Despite the LTLS originally having been used to provide a platform for industrial engineering education by improving the process and operation performance, this work offered the possibility of integrating other engineering disciplines in order to learn within a lean manufacturing context. Hence, the DeLTLS offered engineering students from disciplines such as robotics, mechatronics, or information technology engineering a practical, real-world scenario to increase their Industry 4.0 knowledge and skills in line with Education 4.0. Thus, the learning results have a wider vision and scope than those involving a single engineering discipline alone.

Concerning the research criteria of reliability, transferability, and validity, the interpretation of the results suggests the following. First, we can claim the reliability of the research results of the students' competency assessments, final project grades, and course opinions, which were consistent in indicating the contribution of the learning space to the enhancement of their learning relevance, interest, and motivation. Hence, all the data point consistently to the contributions of the learning space. Second, this work offers a framework, method, and assumptions and illustrates their use to develop challenging learning experiences in the DeELS. Thus, other researchers can use and replicate this work to create new instances of learning experiences and study their value. Finally, this work provides two application cases of learning experiences implemented as part of undergraduate engineering courses. All the data collected in each learning experience suggest the internal validity of the two individual cases. The collected data can only refer to the individual case study, and no inferences or generalizations can be made regarding other instances of learning spaces or learning experiences. Further instances of the learning space and learning experiences are required in order to conduct deeper analyses and identify relationships or patterns. Thus, the external validity of these results is limited.

### 4.2. Limitations and Future Work

The limitations of this work refer to the use of the pedagogical approach, the case-study method applied in the methodology, the data collection process, and the execution of the learning experiences.

First, the use of experiential learning as the underpinning pedagogical approach raises some arguments about the use of this approach and the way in which learning is conceptualized. As the experiential learning cycle involves four stages of learning, the learning activities have to be organized accordingly, following the recommended sequence of concrete experience, reflective observation, abstract conceptualization, and active experimentation. Despite the lack of negative comments, feedback, or reactions obtained from the students, the design of the learning activities was demanding and time-consuming for the instructors. Experiential learning demands a well-structured set of activities in alignment with the learning outcomes, learning competencies, learning challenges, and the recreated learning space. However, this might be seen as a strength in pedagogical terms.

Second, the use of a case study method aided in the reporting of the learning experiences; however, the external validity of the results is limited. Our findings can only apply to the corresponding learning experience, and future work is required to identify invariances or similarities in order to draw inferences or generalizations about DeELS and their contributions to the development of learning outcomes and digital competency.

Third, regarding the data collection process, the students' opinions were collected using anonymous and voluntary surveys, which impeded the coding of the data and the collection of answers from all the students. Hence, the numeric data on the students' marks are the only data covering all the students who participated in the learning experiences.

Finally, the execution of one of the learning experiences was disrupted by the COVID-19 pandemic, which forced the tutors to conduct it remotely. In this sense, the recreated learning space itself was different from the face-to-face version of the LTLS, despite the learning challenges and learning activities being similar. Therefore, this opens up possibilities for the recreation of other types of experiential learning spaces in the future.

However, this work offers a framework and method that can be used to develop DeELS and produce student learning outcomes and digital competency, a research methodology to guide future work, and research instruments and statistical methods for the study and interpretation of results.

Concerning future work, there is potential to develop new learning experiences, opening up the alternative of validating the impact of the DeELS on the development of engineering learning outcomes and digital competency in Education 4.0. The further transformation of the DeLTLS could unfold in two possible types of learning experiences:

(i) the students transforming the learning space by incorporating other digital technologies (such as those reported in this manuscript), and (ii) the students operating in the newly transformed learning space using the implemented digital technologies to improve the process performance and other production metrics. There is also the possibility of replicating the proposed framework and its method in other disciplines, learning spaces, and contexts or scenarios (e.g., countries, universities, schools, stakeholders, and situations). Additionally, there is also potential to develop other engineering learning outcomes and digital competencies in other DeELS, considering other types of processes or operations in different manufacturing, production, or other service-oriented experiential learning spaces.

Further work is also required in order to develop and assess additional learning outcomes, report grades, and instructors' and students' opinions, aiming, in particular, to obtain more data results and analyze similarities and differences between different implementations and variable correlations. This proposition could help us to overcome the current limitation of this work based on the use of case studies. All these elements may also help other researchers to replicate and use this work. For now, this work cannot offer wider generalizations or claims about the contribution of this DeELS to the development of digital learning outcomes among students.

## 5. Conclusions

This work advanced the conceptualization and exploration of DeELS to develop challenging learning experiences concerning digital transformation requirements, following the experiential learning cycle for Education 4.0 in engineering education. The results suggest that DeELS enable the development of engineering learning outcomes and digital competency in this type of educational setting. This proposition establishes a novel idea of learning spaces beyond rooms and the traditional physical infrastructure of laboratories. Its contribution to educational practice includes a conceptualization, framework, and method that can be applied to create new instances of the DeELS and relevant learning experiences of Education 4.0 in engineering education.

This work presented two application cases carried out within the LTLS: The *digital kanban system and the smart double-bin kanban*. The students solved learning experience challenges to improve the flow and replenishment of materials using smart sensors, cloud computing, wearables, and other tools. These experiences helped to enrich an innovative experiential learning space with digital technologies, resulting in the digitally enabled LTLS.

However, there is a need to continue exploring these variables related to the level of achievement in competency development, instruments, and techniques in order to observe and evaluate disciplinary and sustainability competencies. The application cases presented here suggest that future work is required in order to provide wider inferences or generalizations in other instances about the contribution of DeELS to the development of digital competencies and to enhance the motivation, interest, and engagement of students. The application cases only present findings limited to the two implementations.

Moreover, the methodology helped us to develop and evaluate the implementation of learning experiences in the DeELS for Education 4.0 in engineering education. The application cases incorporated new technologies as challenging scenarios in order to conduct experiments and define the best technology-enhanced Education 4.0 learning experiences. Furthermore, this work could pave the way for new learning spaces, experiences, and educational methods for new practices that could impact educational public policies concerning Education 4.0 at the country level in the years to come.

This work also contributes to and aims to support future work, as new experiential learning instances can be envisioned for digital transformation and Industry 4.0 technologies in the learning process of Education 4.0. This future work will invite other engineering programs or disciplines to collaborate in novel learning experience challenges that increase digitalization knowledge. The DeELS may be used as a multi-, inter-, or transdisciplinary

general learning space for the promotion of the different types of capacities, abilities, and skills, either in higher education or lifelong learning.

**Author Contributions:** Conceptualization, D.E.S.-N., C.L.G.-R. and I.A.A.-S.; methodology, D.E.S.-N. and C.L.G.-R.; validation, D.E.S.-N., C.L.G.-R. and I.A.A.-S.; formal analysis, D.E.S.-N. and C.L.G.-R.; investigation, D.E.S.-N., C.L.G.-R. and I.A.A.-S.; resources, D.E.S.-N., C.L.G.-R. and I.A.A.-S.; data curation, D.E.S.-N., C.L.G.-R. and I.A.A.-S.; writing—original draft preparation, D.E.S.-N., C.L.G.-R. and I.A.A.-S.; writing—review and editing, D.E.S.-N., C.L.G.-R. and I.A.A.-S.; visualization, D.E.S.-N., C.L.G.-R. and I.A.A.-S.; supervision, D.E.S.-N.; project administration, D.E.S.-N. and C.L.G.-R.; funding acquisition, D.E.S.-N.; C.L.G.-R. and I.A.A.-S. All authors have read and agreed to the published version of the manuscript.

**Funding:** The APC was funded by the Writing Lab, Institute for the Future of Education, Tecnologico de Monterrey, Mexico.

**Institutional Review Board Statement:** Ethical review and approval were waived for this study because the review board deemed it "research without risk,", i.e., studies using retrospective documentary research techniques and methods, as well as those that do not involve any intervention or intended modification of the physiological, psychological, and social variables of the study participants, among which the following are considered: questionnaires, interviews, review of clinical records, and others, in which the participants are not identified, nor are sensitive aspects of their behavior addressed.

**Informed Consent Statement:** Not applicable.

**Data Availability Statement:** The data presented in this study are available on request from the corresponding authors (D.E.S.-N. and C.L.G.-R.).

**Acknowledgments:** The authors wish to acknowledge the financial and technical support of Writing Lab, Institute for the Future of Education, Tecnológico de Monterrey, Mexico, in the production of this work.

**Conflicts of Interest:** The authors declare no conflict of interest. The funders had no role in the design of the study, in the collection, analysis, or interpretation of data, in the writing of the manuscript, or in the decision to publish the results.

## Appendix A. Digital Transformation Competency Rubrics

**Digital Transformation Competency** Definition: The student generates solutions to problems in the professional field through the intelligent and timely incorporation of novel digital technologies.

**Sub-Competency Cutting-Edge Technologies:** The student evaluates various technologies with openness in order to search for and implement relevant alternatives for the transformation of professional practice according to economic, environmental, social, political, ethical, safety and hygiene, and manufacturability restrictions.

**Table A1.** The rubric of the observed sub-competency.

| LEVEL A: DESIRABLE | LEVEL B: ACCEPTABLE | LEVEL C: INSUFFICIENT |
|---|---|---|
| - Knows, evaluates, and selects the relevant Industry 4.0 technologies for digital transformation in professional practice.<br>- Designs prototypes and their operations or functions in manufacturing or services.<br>- Awareness of the importance of digital transformation according to economic, environmental, social, political, ethical, safety and hygiene, and manufacturing restrictions.<br>- Applies engineering techniques and tools to solve problems in the context of digital transformation. | - Knows and selects Industry 4.0 technologies for digital transformation in professional practice.<br>- Designs prototypes with some deficiencies in their operation or functioning in the field of manufacturing or services.<br>- Shows an awareness of the importance of digital transformation, without considering one or more of the economic, environmental, social, political, ethical, safety and hygiene, and manufacturing restrictions.<br>- Applies some engineering techniques and tools to solve problems. | - Knows or selects Industry 4.0 technologies for digital transformation in professional practice.<br>- Proposes prototypes with deficiencies in operation or functioning.<br>- There is no evidence of the importance of digital transformation regarding economic, environmental, social, political, ethical, health and safety, and manufacturing restrictions.<br>- Loosely applies some engineering techniques and tools to solve problems. |

**Table A2.** Single-point checklist for digital competency.

| Aspect to Improve: | Not Observed (Yes/No) | Observable Criterion: | Observed (Yes/No) | Highlights: |
|---|---|---|---|---|
| | | Knows about various relevant technologies for professional practice. | | |
| | | Evaluates the various technologies of Industry 4.0 and selects the relevant one(s) to solve problems or improve systems in professional practice. | | |
| | | Uses engineering techniques and tools to solve problems in the context of digital transformation. | | |
| | | Shows the design of a prototype and its operation or functioning in real manufacturing situations or services. Or shows the design of a prototype and its operation with the support of simulators. | | |
| | | Shows an awareness of the importance of digital transformation according to economic, environmental, social, political, ethical, safety and hygiene, and manufacturing restrictions. | | |

## Appendix B. Final Project Report Evaluation Rubric

**Table A3.** Project teamwork observation guide: developing an Industry 4.0 technology enabler.

| Evaluation Criteria (Weight) | | Meets Expectations (100–81%) | Sufficiently Meets Expectations (80–51%) | Does Not Meet Expectations (50–0%) | Comments Section |
|---|---|---|---|---|---|
| Project Objectives (10 points) | The objectives of the laboratory project are precisely defined, as is its scope. | | | | |
| Kanban System (10 points) | Students clearly explain what a kanban system is and how important it is in the industry. | | | | |
| Prototype General Description (10 points) | The rendered image of the designed prototype is shown, and its operation or functioning is described through phases or stages | | | | |
| Mechanical Design (10 points) | The following CAD parts are presented: <br> • Exploded view of the entire assembly (with its respective parts in a table). <br> • Drawings (with dimensions) of the main parts. | | | | |
| Selection of Actuators and Sensors (10 points) | Based on the above information, the prototype actuators and sensors' selection are justified, showing comparative tables comparing different options for said actuators and sensors. | | | | |
| Prototype Automation (20 points) | Boolean functions and/or phase-space diagrams that define the control logic of the prototype are clearly explained. | | | | |
| Simulation (20 points) | Students simulate their prototype's logic, which faithfully represents the described behavior and the Boolean functions and/or the defined phase-space diagrams. | | | | |
| Conclusions (10 points) | The general conclusions of the project are presented, and the recommendations for its implementation in future semesters are provided. | | | | |

## Appendix C. Survey of Learning Experiences for Digital Transformation of the LTLS

We appreciate your participation in this questionnaire for educational innovation in regard to digital transformation issues!

INSTRUCTIONS: Read carefully and select the option that best suits YOUR EXPERIENCE in designing the digital kanban system and smart double-bin kanban.

Questions:

- How RELEVANT for the professional practice of your discipline was developing and/or implementing any Industry 4.0 technologies during the kanban system challenge?

5. Highly relevant.     4. Relevant.     3. Moderately relevant.
2. Slightly relevant.     1. Not relevant.

- What level of MOTIVATION in regard to your learning did you experience through the challenge of the kanban system?

  5. Very high motivation.  4. High motivation.  3. Medium motivation.
  2. Low motivation.  1. No motivation.

- What level of INTEREST did this project generate in you to learn about Industry 4.0 technologies in your future professional practice?

  5. Very high interest.  4. High interest.  3. Medium interest.
  2. Low interest.  1. No interest.

- How do you assess the development of your "DIGITAL TRANSFORMATION" competency as you designed and/or implemented engineering solutions using Industry 4.0 enablers in regard to economic, environmental, social, political, ethical, security, hygiene, and manufacturing issues?

  5. Very good development.  4. Good development.  3. Medium development.
  2. Little development.  1. No development.

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
