# Peer review of "Digitally Enabled Experiential Learning Spaces for Engineering Education 4.0"

_education, doi:10.3390/educsci13010063_

Round 1

Reviewer 1 Report

The present study proposed an approach for Digitally-enabled Experiential Learning Spaces (DeELS) by incorporating digital technologies in the context of engineering education. In a time when the education sector is already too familiar with technology-driven strategies, this paper could contribute a useful approach to the new era of modern education. I find the paper acceptable but the authors may consider the following:

• The paper is rather too long and it is difficult to keep engaged reading the whole paper. It gives me the impression that there are too many pieces that should have been separated. For example, the authors stated that they conducted a systematic literature review to know the state-of-the-art learning spaces. The extensiveness of this review is an entire paper but was included in the paper with not many details. 

• This review was also not contextualized in engineering education. Was there a reason for this?

• Despite the many concepts in the paper, no discussion was offered on the application of experiential learning in engineering education. There are many pedagogies available that may establish the underpinning of experiential learning. For instance, hackathons as extracurricular activities (DOI: 10.1002/cae.22564) and other design competitions (DOI: 10.1177/030641901774958) in engineering education are becoming more popular as a form of experiential learning.

• Digitally-enabled Experiential Learning Spaces framework should be labeled figure 3.

Section 3 Results do not report any results. The results were rather presented in the Section 4 Discussion. The authors may rearrange the structure of the paper. The discussion section should discuss the implications of findings based on the previous literature.

• There is no Table 9. It skipped directly to Table 10.

• The authors may acknowledge the limitations of the study.

Author Response

Dear Reviewer 1,

We much appreciate the comments and suggestions, which have been very helpful in improving our work. We have carefully studied the comments and made significant corrections to comply with your expectations.

We hope the revised manuscript will suit the standards of the Journal, and we thank you for your interest in our work.

Attached you can find our responses.

Reviewer 2 Report

Comments can be found in the attached PDF.

Author Response

Dear Reviewer 2,

We much appreciate the comments and suggestions, which have been very helpful in improving our work. We have carefully studied the comments and made significant corrections to comply with your expectations.

We hope the revised manuscript will suit the standards of the Journal, and we thank you for your interest in our work.

Attached you can find our responses.

Reviewer 3 Report

The paper is well -written and the authors talked about new teaching direction related to industrial and system engineering and new Digitally-enabled Experiential Learning Spaces. The needs and trends are clearly presented. 

Authors discussed about Education 4.0 . However, it is not quite sure the evolution in education aspect or authors mention educate student with Industry 4.0 concept. 

Figure 1 and Figure 5 are highly similar and it is better to use one Figure. 

The paper mentioned about Challenge-based Learning experience and the student project made used of mechantroic and Industrial engineering concept to design a digital kanban system. It is quite impressive. Maybe more information about the number of groups of students involved should be stated in the manuscript. For those project, the academic weak students may have difficulties . How do they contribute and overcome the challenge? 

For the appendix, the detail evaluation and rubric are included and it is good template for industry 4.0 challenge based project. 

Author Response

Dear Reviewer 3,

We much appreciate the comments and suggestions, which have been very helpful in improving our work. We have carefully studied the comments and made significant corrections to comply with your expectations.

We hope the revised manuscript will suit the standards of the Journal, and we thank you for your interest in our work.

Attached you can find our responses.

Reviewer 4 Report

Dear authors, thanks for sharing this type of paper with us. The topic addressed is relevant and of interest to the scientific community. However, in my opinion, some sections must be improved in order to be considered for publication in this journal.

About technical aspects:

The title does not reflect the article's content; the title is ambiguous and should contain more elements that describe the field of development and the case studies presented in a particular way. It is not appropriate to generalize it as currently stated. The article shows two cases of implementation within the field of industrial engineering. The title should reflect this. In my opinion, the authors should adjust the title.

Since many well-known concepts jump in this article, it is necessary to show clearly to what extent this work contributes to knowledge and why it is a novelty. This work looks like a typical case study article implementing technology in an educational context. Please, you should try to clarify this point.

Learning experiences in an Education 4.0 context vary and depend on the context and set of teaching-learning dynamics, stakeholders, infrastructure, and technologies addressed. From my point of view, how a learning space and learning experiences are described in the context of Education 4.0 in this work are limited and ambiguous, and it is necessary to complement and detail. Also, a set of specific competencies are related to Education 4.0, and it is unclear how the addressed competencies in this work relate to Education 4.0. The competencies you mention seem typical competencies without the context of Education 4.0. What competencies are desirable to measure in these case studies that you present?

A core topic is “digitally-enabled experiential learning spaces” in my point of view; this topic should be addressed in a more particular and detailed way. Education 4.0 is more than this. What is and what is not a Digitally-enabled experiential learning spaces in an Education 4.0 context? What background is around these issues? Why is it new? What research question do you have?

For me, the stated objectives are not clear. It even seems that they have not been adequately included in this work: “Therefore, this work aims to (i) offer a framework to develop education 4.0 learning experiences, (ii) a method to implement this framework, and (iii) exemplify and report the use of these ideas in undergraduate engineering courses.”

There is the reference framework design theory and its design and creation process are not reflected in this work. A Framework has standard elements that are not reflected in this work. This work shows models of case studies, not a framework.

The work addresses Engineering Education in a very general way. However, it focuses on Industrial Engineering, leaving the subject of Engineering Education very ambiguous.

About format aspects:

Your figures present really bad quality, including pictures (photos). You have to review this.

Education 4.0 is mentioned using upper “E” and other times using lower “e.” Please note this. “Education 4.0” “education 4.0”

Contains a large number of citations. And many of them are from years of more than five years. It is recommended to make the necessary adjustments to considerably reduce the number of citations and find this work with more current references.

Author Response

Dear Reviewer 4,

We much appreciate the comments and suggestions, which have been very helpful in improving our work. We have carefully studied the comments and made significant corrections to comply with your expectations.

We hope the revised manuscript will suit the standards of the Journal, and we thank you for your interest in our work.

Attached you can find our responses.

Round 2

Reviewer 4 Report

Thanks to consider the provided comments